# Statistical modelling of co-seismic knickpoint formation and river response to fault slip

Philippe Steer[1], Thomas Croissant[1,a], Edwin Baynes[1,b], Dimitri Lague[1]

[1]Univ Rennes, CNRS, Géosciences Rennes - UMR 6118, F-35000 Rennes, France.
[a]now at Department of Geography, Durham University, Durham, UK.
[b]now at Department of Civil and Environmental Engineering, University of Auckland, Auckland, New Zealand

*Correspondence to*: Philippe Steer (philippe.steer@univ-rennes1.fr)

**Abstract.** Most landscape evolution models adopt the paradigm of constant and uniform uplift. It results that the role of fault activity and earthquakes on landscape building is understood under simplistic boundary conditions. Here, we develop a numerical model to investigate river profile development subjected to fault displacement by earthquakes and erosion. The model generates earthquakes, including mainshocks and aftershocks, that respect the classical scaling laws observed for earthquakes. The distribution of seismic and aseismic slip can be partitioned following a spatial distribution of mainshocks along the fault plane. Slope patches, such as knickpoints, induced by fault slip are then migrated at a constant rate upstream a river crossing the fault. A major result is that this new model predicts a uniform distribution of earthquake magnitude rupturing a river that crosses a fault trace and in turn a negative exponential distribution of knickpoint height for a fully coupled fault, i.e. with only co-seismic slip. Increasing aseismic slip at shallow depths, and decreasing shallow seismicity, censors the magnitude range of earthquakes cutting the river towards large magnitudes and leads to less frequent but higher amplitude knickpoints, on average. Inter-knickpoint distance or time between successive knickpoints follows an exponential decay law. Using classical rates for fault slip, 15 mm.yr$^{-1}$ and knickpoint retreat, 0.1 m.yr$^{-1}$, leads to high spatial densities of knickpoints. We find that knickpoint detectability, relatively to the resolution of topographic data, decreases with river slope that is equal to the ratio between fault slip rate and knickpoint retreat rate. Vertical detectability is only defined by the precision of the topographic data that sets the lower magnitude leading to a discernible offset. Considering a retreat rate with a dependency on knickpoint height leads to the merging of small knickpoints into larger one and larger than the maximum offset produced by individual earthquakes. Moreover, considering simple scenarios of fault burial by intermittent sediment cover, driven by climatic changes or linked to earthquake occurrence, leads to knickpoint distributions and river profiles markedly different from the case with no sediment cover. This highlights the potential role of sediments in modulating and potentially altering the expression of tectonic activity in river profiles and surface topography. The correlation between the topographic profiles of successive parallel rivers cutting the fault remains positive for distance along the fault of less than half the maximum earthquake rupture length. This suggests that river topography can be used for paleo-seismological analysis and to assess fault slip partitioning between aseismic and seismic slip. Last, the developed model can be coupled to more sophisticated landscape evolution models to investigate the role of earthquakes on landscape dynamics.

## 1 Introduction

The interactions among tectonics, climate and surface processes govern the evolution of the Earth's topography (e.g. Willet et al., 1999; Whipple, 2009). Among the potential link and feedbacks between tectonics and surface processes, the building of topographic slopes by tectonic deformation is critical. Erosion rates and most geomorphological processes are strongly sensitive to local slope, including river incision (e.g. Whipple and Tucker, 1999), glacial carving (e.g. Herman and Braun, 2008), soil creep (e.g. McKean et al., 1993) and hillslope mass wasting (e.g. Keefer, 1994). The dependency to slope can be linear or non-linear, mainly due to threshold effects or to a power-law behaviour. For instance, a theoretical model combined with a data compilation suggests that river incision rate is linearly dependent on slope at knickpoints, and more than linearly dependent on slope for more gentle stream profiles (Lague, 2014). This is pivotal, as temporal variations in tectonic displacement and in slope building cannot be averaged out when considering river profile evolution using an erosion law with a non-linear dependency to slope. In addition to slope, the height of knickpoints (i.e. with a slope above average local slope) and waterfalls (i.e. with a slope close to infinity) appears as a fundamental ingredient of their survival, retreat rate and river incision (Hayakawa and Matsukura, 2003; Baynes et al., 2015; Scheingross and Lamb, 2017). Similar issues arise for hillslope dynamics impacted by fault scarp development (Arrowsmith et al., 1996) and possibly for faults in glaciated landscapes. Despite this, most landscape evolution models of topographic growth consider slope building as a continuous process resulting from a constant (or smoothly varying) uplift rate (e.g. Braun and Willett, 2013; Thieulot et al., 2014; Campforts et al., 2017). There is therefore a clear need to define how tectonic deformation builds topographic slopes in numerical models.

The expression of tectonic deformation on topographic slope is diverse, and its spatial and temporal scales range from meters to continents and from instantaneous to geological times, respectively. Tectonic deformation can 1) instantaneously generate steep-to-infinite slopes when earthquakes rupture the Earth's surface (e.g. Wells and Coppersmith, 1994); 2) induce progressive slope building at the orogen scale and over a seismic cycle by aseismic deformation (i.e. deformation not associated to earthquakes) and interseimic deformation (i.e. deformation occurring in-between large magnitude earthquakes) (e.g. Cattin and Avouac, 2000) or by the deformation associated to earthquakes with no surface rupture; and 3) lead to longer-term topographic tilting at the orogen-to-continental-scale by isostatic readjustment (e.g. Watts, 2001) or viscous mantle flow (e.g. Braun, 2010). In this paper, we focus on the building of topographic slopes by fault slip at the intersection between a fault trace and a river. This is motivated first by the fact that the greatest slopes are expected to occur by faulting, and second by the already well understood role of isostasy and viscous deformation on topography (e.g. Watts, 2001; Braun, 2010). In active mountain belts, displacement along frontal thrust faults can lead to the development of co-seismic waterfalls, knickpoints and knickzones than can reach several meters of elevation (e.g. Boulton and Whittaker, 2009; Yanites et al., 2010; Cook et al., 2013). These differential topographies, associated to high slopes, are referred to as slope patches in the following work. These slope patches have long been recognized as potential markers of the dynamic response of rivers (e.g. Gilbert, 1896) to transient conditions, not limited to changes in tectonic activity, and including base level fall and lithological contrasts, among others. Yet, in active tectonic areas, knickpoints are frequently associated to fault activity and transience in uplift rate (e.g. van der

Beek et al., 2001; Quigley et al., 2006; Dorsey and Roering, 2006; Yildirim et al., 2011). These slope patches generated by frontal thrusts along a river migrate upstream by erosion and are expected to set the erosion rate of the entire landscape (Rosenbloom and Anderson, 1994; Royden and Perron, 2013; Yanites et al., 2010; Cook et al., 2013).

Fault slip and surface rupture classically occur by seismic slip during earthquakes. However, associating individual earthquakes to knickpoints, or associating series of knickpoints to series of earthquakes remains challenging from field data. We therefore use in this paper a statistical model of earthquakes to simulate the expected slope and height distributions of the slope patches generated by earthquakes (i.e. fault seismic slip) and fault aseismic slip at the intersection between a thrust fault and a river. This model uses the branching aftershock sequence (BASS) model (Turcotte et al., 2007) to simulate temporal and spatial series of earthquakes based on the main statistical and scaling laws of earthquakes. The rupture extent and displacement of earthquakes are inferred using classical scaling laws (Leonard, 2010). We focus on the response of rivers and analyse the resulting knickpoint height distribution and their migration distance along a single river, in near-fault conditions. We also infer the correlation between the topography of successive parallel rivers distributed along the strike of a single fault. The obtained results are then discussed with regards to the potential of knickpoints and waterfalls to offer paleo-seismological constraints, and to the necessity of considering time-variable uplift accounting for earthquake sequences in landscape evolution models. It is important to stress out that this study does not aim to investigate specific geomorphological settings, but to give general theoretical and modelling arguments to the interpretation of river profiles upstream of active faults.

## 2 State of the art: linking fault slip to knickpoint formation and migration

### 2.1 From fault slip and earthquakes to surface ruptures and knickpoints

In near fault conditions, too few data characterizing fault rupture geometry at one location (e.g. along a river) exist to assess the distribution of the slope and height of surface ruptures resulting from earthquakes by local fault activity (e.g. Ewiak et al., 2015; Wei et al., 2015; Sun et al., 2016). Regional or global compilation of fault rupture by earthquakes (e.g. Wells and Coppersmith, 1994; Leonard, 2010; Boncio et al., 2018) offer another approach, that yet suffers from inescapable statistical biases mainly due to the use of faults with different slip rates, dimensions, seismogenic properties and records of paleo-earthquakes. In addition, small earthquakes associated with small rupture extents and co-seismic displacement are less likely to be identified in the field. For instance, using seismological scaling laws (Leonard, 2010), an earthquake of magnitude 3 on a thrust fault has a rupture length of 188 m and an average displacement of 1.2 cm. This displacement is clearly below the precision of current digital elevation models or in any case hidden by the inherent topographic roughness.

Statistical or theoretical inferences offer another means to associate fault activity and earthquakes to surface ruptures and knickpoints. Earthquakes tend to universally follow the Gutenberg-Richter frequency-magnitude distribution in Eq. (1):

$$\log_{10}\big(N(\geq Mw)\big) = a - bMw, \tag{1}$$

where $Mw$ is the magnitude, $N(\geq Mw)$ is the number of earthquakes with magnitudes greater or equal to $Mw$, $b$ is the exponent of the tail (referred to as the b-value), generally observed to be close to 1 ($0.5<b<1.5$), and $a$ characterizes earthquake productivity (Gutenberg and Richter, 1944). The definitions of all variables used in this paper are summarized in Table C1. Assuming self-similarity, a b-value of 1 can be interpreted as the result of the successive segmentation of larger earthquakes into smaller earthquakes (Aki, 1981; King, 1983) so that any point along a 2D fault plane, including the intersection between the fault trace and a river, displays a uniform probability to be ruptured by earthquakes of any magnitude. This inference only stands if the distribution of earthquakes along the fault plane is uniform. However, fault slip can occur by seismic slip, but also by aseismic deformation, including interseismic creep, postseismic deformation and slow slip events (e.g. Scholz, 1998; Peng and Gomberg, 2010; Avouac, 2015). The relative spatial and temporal distribution of aseismic and seismic slip along a fault plane is variable and still poorly understood. Yet, experimental results and the depth distribution of earthquakes along subduction or intraplate thrust faults suggest that shallow depths (< 5 km) are favourable to frictional stability and in turn to aseismic slip (Scholz 1998). This probably censors the magnitude range of earthquakes rupturing the surface towards large magnitudes associated with rupture extent greater than this minimum seismogenic depth.

## 2.2 Knickpoint formation

The transformation of surface ruptures into knickpoints remains a relatively enigmatic issue. Linking knickpoints to individual earthquakes is challenging, although some recently formed knickpoints have been clearly identified as the result from the surface rupture of a single large earthquake (e.g. Yanites et al., 2010; Cook et al., 2013). The transformation of individual surface ruptures into individual co-seismic knickpoints is not necessarily a bijective function and is more likely to be a surjective function. In other words, a knickpoint can be made of several surface ruptures. Indeed, if the time interval between two (or more) successive ruptures at the same location is less than a characteristic migration time required to segregate their topographic expressions, then the formed knickpoint will result from this succession of surface ruptures and earthquakes. An end-member setting favouring this behaviour is the case of fault scarps developing on hillslopes, which degradation is generally assumed to follow a diffusion law (e.g. Nash, 1980; Avouac, 1993; Arrowsmith et al., 1996; Roering et al., 1999; Tucker and Bradley, 2010). Moreover, in the downstream part of rivers, fault scarps can remain buried under a sediment cover due, for instance, to the development of an alluvial fan (Finnegan and Balco, 2013; Malatesta and Lamb, 2018). Development of the fault scarp height by successive ruptures or the thinning of the alluvial cover can then expose the scarp, in turn potentially forming a knickpoint that can erode and migrate. This intermittent fault burial mechanism can therefore produce knickpoints formed by the surface rupture of several earthquakes.

The burial of the fault during successions of aggradation-incision phases of an alluvial fan located immediately downstream of the fault (e.g. Carretier and Lucazeau, 2005) has not been considered in previous landscape evolution models. This mechanism is suggested to be a primary control of knickpoints and waterfalls formation by allowing the merging of several small co-seismic scarps formed during burial phases into single high-elevation waterfalls that migrate during latter incision phases (Finnegan and Balco, 2013; Malatesta and Lamb, 2018).

## 2.3 Knickpoint migration and preservation

Once formed, knickpoints can migrate upstream due to river erosion. Over geological time-scales ($> 10^3$ yr), rates of knickpoint retreat for bedrock rivers typically range between ~$10^{-3}$ and ~$10^{-1}$ m.yr$^{-1}$ (e.g. Van Heijst and Postma, 2001). This range is also consistent with the order of magnitude of documented knickpoint retreat rates in Eastern Scotland (Bishop et al., 2005; Jansen et al., 2011), around ~$10^{-1}$ m.yr$^{-1}$, in the Central Apennines, Italy and in the Hatay Graben, Southern Turkey (Whittaker and Boulton, 2012), between ~$10^{-3}$ and ~$10^{-2}$ m.yr$^{-1}$. However, on shorter time scales, significantly higher rates can be found with values potentially reaching ~$10^{0}$ or even ~$10^{1}$ m.yr$^{-1}$. For instance, the Niagara Falls retreated at a rate of a few meters per year over tens of years (Gilbert, 1907) and some knickpoints formed by the 1999 Chi-Chi earthquake in Taiwan even retreated by a few hundreds of meters over about ten years (Yanites et al., 2010; Cook et al., 2013). A more extensive analysis of the range of knickpoint retreat rates in relation to the observation time-scale can be found in Van Heijst and Postma (2001) and in Loget and Van Den Driessche (2009).

In detachment-limited conditions, the stream power incision model predicts that knickpoint horizontal migration or retreat follows a linear or non-linear kinematic wave in the upstream direction, depending on the slope exponent (e.g. Rosenbloom and Anderson, 1994; Tucker and Whipple, 2002; Whittaker and Boulton, 2012; Royden and Perron, 2013). This prediction is supported by the apparent correlation between retreat rate and drainage area or water discharge, deduced from field observation and experimental studies (Parker, 1977; Schumm et al., 1987; Rosenbloom and Anderson, 1994; Bishop et al., 2005; Crosby and Whipple, 2006; Loget et al., 2006; Berlin and Anderson, 2007). However, some experimental results show no dependency of retreat rate on water discharge, (Holland and Pickup, 1976), possibly due to the self-regulatory response of river geometry to water discharge through change in river channel width (Baynes et al., 2018). Other factors influencing retreat rate include, among others, sediment discharge (e.g. Jansen et al., 2011; Cook et al., 2013), flood events (e.g. Baynes et al., 2015), rock strength (e.g Stock and Montgomery, 1999; Hayakawa and Matsukura, 2003; Baynes et al., 2018), fracture density and orientation (Anton et al., 2015; Brocard et al., 2016) and the spacing and height of the waterfalls (Scheingross and Lamb, 2017).

Preservation of knickpoint shape during retreat is poorly understood as very little data exist on the temporal evolution of their shape. For instance, knickpoints along the Atacama Fault System are systematically reduced in height compared to the height of ruptures directly on the fault scarp (Ewiak et al., 2015). At the opposite, ten years after Chi-Chi earthquake, the height of co-seismic knickpoints was ranging from 1 to 18 m (Yanites et al., 2010), while the initial surface rupture was limited to 0.5 to 8 m in height (Chen et al., 2001). Theoretically, only the stream power model with a linear dependency on slope predicts the preservation of knickpoint shape, favoured by a parallel retreat (e.g. Rosenbloom and Anderson, 1994; Tucker and Whipple, 2002; Royden and Perron, 2013). A less than linear dependency on slope leads to concave knickpoints, while a more than linear dependency on slope leads to convex knickpoints. Transport-limited models, that reduce to advection-diffusion laws, lead to a diffusion of the differential topography associated to knickpoints. However, transport-limited models are likely more pertinent to predict the evolution of fault scarps along hillslopes (e.g. Rosenbloom and Anderson, 1994; Arrowsmith et

al., 1996; Arrowsmith et al., 1998; Tucker and Whipple, 2002), and evidence points toward a linear dependency on slope for knickpoint erosion (Lague, 2014). Yet, the transformation of fault activity and slip during earthquakes to knickpoints and hillslope scarps and their preservation throughout their subsequent erosion and retreat remains a challenging issue.

## 3 Methods

### 3.1 Fault setting

The tectonic setting considered here is the one of a typical active intracontinental thrust fault, able to generate earthquakes up to magnitude 7.3. The thrust fault has a length $L = 200$ km, a width $W = 30$ km, and a dip angle $\theta = 30°$ so that the fault tip is located at 15 km of depth. The duration of the simulation $T$ is set to 10 kyr to cover many seismic cycles and for earthquakes to be well distributed along the finite fault plane. A schematic sketch illustrates the model setup (Fig. 1).

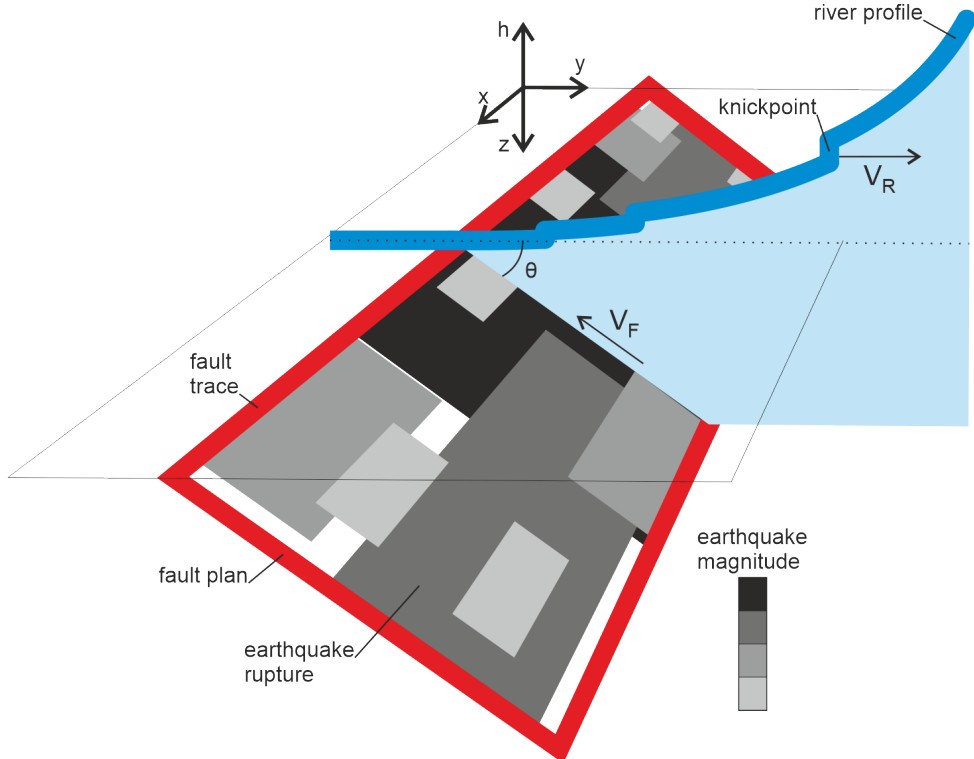

**Figure 1.** Schematic sketch showing the model setup. The fault plan, dipping with an angle $\theta$, is represented by a red contour and includes the earthquakes and their ruptures represented by gray box, which color indicates the magnitude. The fault trace is aligned along the $x$-axis and earthquakes occur at depth $z$. The river profile is indicated by a blue line along the $y$-axis and has an elevation $h$. The river contains several knickpoints. Note that in this paper we only focus on knickpoints occurring in near-fault condition. The rate of fault slip is $V_F$ while knickpoint migrate at a constant velocity $V_R$.

## 3.2 Mainshocks

Mainshocks are generated along the fault plane. The potential magnitude range of mainshocks is bounded by fault width, that set the maximum earthquake rupture width and by a minimum rupture width, here chosen at 500 m. Based on Leonard (2010), the modeled thrust fault allows magnitudes ranging from $Mw_{min} = 3.7$ to $Mw_{max} = 7.3$. Inside these bounds, the magnitude
5  of each mainshock is determined by randomly sampling the Gutenberg-Richter distribution, with a b-value of 1 (Fig. 2). The earthquake productivity of the distribution is inferred based on the arbitrarily chosen rate of mainshock $R = 0.1$ day$^{-1}$, leading to $a = \log_{10}(R\,T) + b\,Mw_{min} = 8.975$. The time occurrence of each mainshock is randomly sampled over the duration of the simulation. Each mainshock is therefore considered independent, and the only relationship between mainshocks is that their population statistically respects the Gutenberg-Richter distribution (Gutenberg and Richter, 1944).

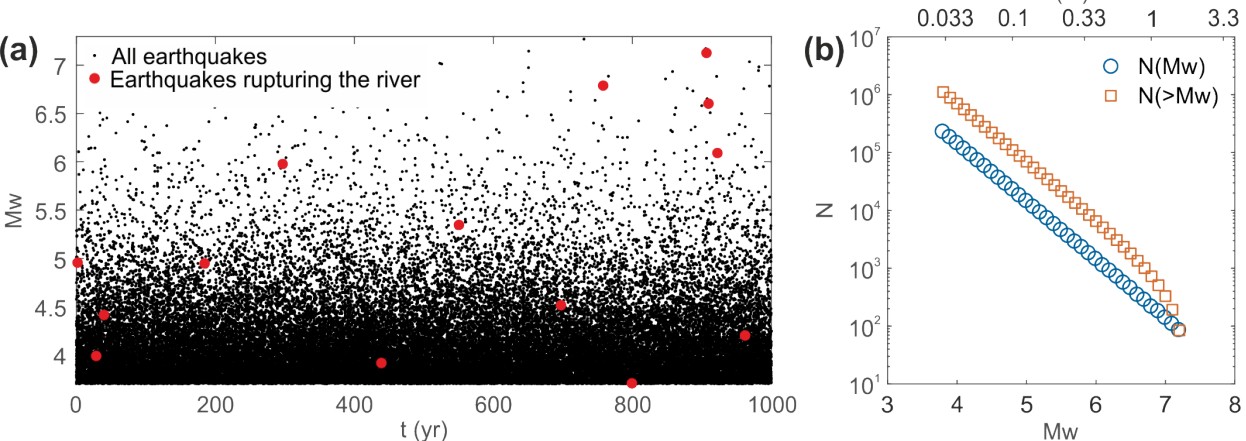

**Figure 2.** Modeled seismicity and its statistical characteristics. a) Time distribution $t$ of the magnitude $Mw$ of earthquakes during the first 1000 years of one model. Both mainshocks and aftershocks are shown with black dots. Earthquakes with rupture zone extending to the surface and cutting the river, located at the middle of the fault trace, are shown with red dots. b) The cumulative (light red squares) and incremental (light blue circles) Gutenberg-Richter magnitude-frequency distribution of earthquakes for one model. $N$ is the number of events
15  and $D$ is the associated displacement computed using Leonard (2010) scaling law.

The spatial location of mainshocks inside the fault plane is sampled using a 2D distribution that correspond to a truncated normal distribution across-strike and to a uniform distribution along-strike (Fig. 3). A normal distribution with depth roughly mimics the depth distribution of natural earthquakes in the upper crust, that tends to show a maximum number of earthquakes
20  at intermediate depth and less towards the top and the tip of the fault (e.g. Sibson, 1982; Scholz, 1998). Therefore, we set the mean of the normal distribution equal to 7.5 km of depth as the fault tip has 15 km of depth so that earthquakes are more numerous at this intermediate depth. We define two end-member models, referred to as 1) the "seismic and aseismic slip" model using a variance of the normal distribution $\sigma = W/10$, corresponding to a narrow depth-distribution, and 2) the "only

seismic slip" model with $\sigma = 3.3W$, corresponding to an almost uniform depth-distribution. We impose that the maximum earthquake frequency, at depth 7.5 km, is equal in-between all the models.

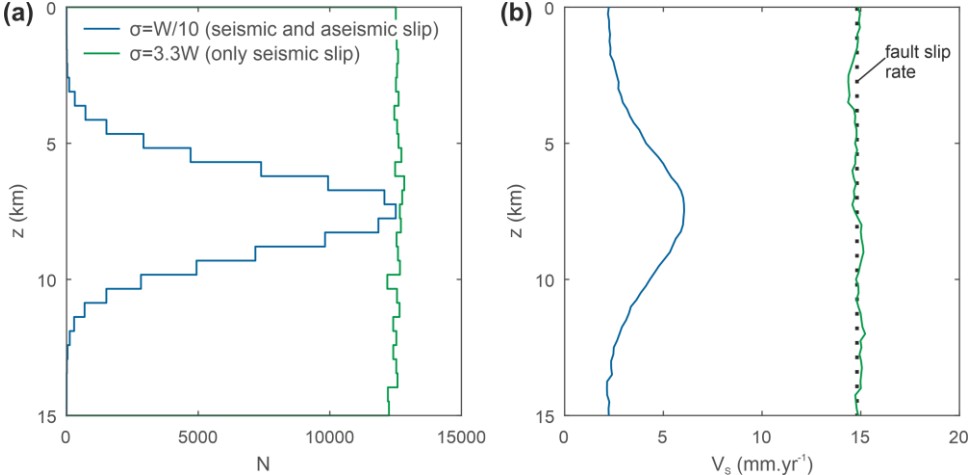

**Figure 3.** Depth distribution of earthquakes, seismic and aseismic slip. a) Depth-distribution of the number $N$ of mainshocks for the 2 models considered here. The depth distribution is a normal one centered at 7.5 km of depth and with a variance $\sigma$ equals to $W/10$ (blue) or $3.3W$ (green). b) Depth-distribution of seismic $V_S$ slip. The vertical black line indicates the averaged fault slip rate of ~15 mm.yr$^{-1}$, summing seismic and aseismic slip. Aseismic slip rate is simply the difference between the average fault slip rate and seismic slip rate, so that all models share the same total slip rate.

### 3.3 Aftershocks

Each mainshock triggers a series of aftershocks that is determined based on the branching aftershock sequence (BASS) model (Turcotte et al., 2007). It represents an alternative to the more classical epidemic type aftershock sequence (ETAS) models (Ogata, 1988), with the advantage of being fully self-similar. We here only briefly describe the BASS model as more details can be found in Turcotte et al. (2007). Based on a mainshock, the BASS model produces a sequence of aftershocks which respect four statistical laws: 1) the Gutenberg-Richter frequency-magnitude distribution (Gutenberg and Richter, 1944; Fig.1); 2) a modified Båth's law (Shcherbakov and Turcotte, 2004), which controls the difference in the magnitude of a mainshock and its largest aftershock; 3) a generalized form of Omori's law describing the temporal decay of the rate of aftershocks (Shcherbakov et al., 2004); and 4) a spatial form of the Omori's law, that controls the spatial distribution of aftersocks (Helmstetter and Sornette, 2003). The BASS model relies on six parameters: the b-value that we set equal to $b = 1$, the magnitude difference $\Delta Mw = 1.25$ of Båth's law, the exponent $p = 1.25$ and offset $c = 0.1$ days of the temporal Omori's law, and the exponent $q = 1.35$ and offset $d = 4.0$ meters of the spatial Omori's law. The values of these aftershock parameters are based on Turcotte et al. (2007) and are constant for all the simulations performed in this paper. Seismicity along the fault is therefore made of mainshocks and their aftershocks. This aftershock model is also similar to the one developed by Croissant et al. (2019).

### 3.4 Earthquake rupture

The length $L_{rup}$, width $W_{rup}$ and average co-seismic displacement $D$ of each earthquake rupture, including mainshocks and aftershocks, are determined using scaling laws with seismic moment $M_O$, empirically determined from a set of intraplate dip-slip earthquakes (Leonard, 2010) following Eq. (2-4):

$$L_{rup} = \left(\frac{M_O}{\mu C_1^{3/2} C_2}\right)^{\frac{2}{3(1+\beta)}}, \tag{2}$$

$$W_{rup} = C_1 L_{rup}^{\beta}, \tag{3}$$

$$D = C_1^{\frac{1}{2}} C_2 L_{rup}^{\frac{1+\beta}{2}}, \tag{4}$$

where $C_1 = 17.5$, $C_2 = 3.8 \ 10^{-5}$ and $\beta = 2/3$ are constants and $\mu = 30$ GPa is the shear modulus (Fig. 1). The location of the rupture patches around each earthquake are positioned randomly to prevent hypocenters being centered inside their rupture patches. The fault has some periodic boundary conditions, in the sense that if the rupture patch of an earthquake exceeds one of the fault limits, the rupture area in excess is continued on the opposite side of this limit. This choice maintains a statistically homogeneous pattern of fault slip rate on the fault plane in the case of the "only seismic slip" model (which displays an almost homogeneous distribution of mainshocks on the fault plane). Another strategy, consisting in relocating each rupture in excess inside the fault limits, was dismissed as it was leading to gradients of fault slip rates close to fault tips.

### 3.5 Seismic and aseismic slip

Slip along the fault plane is partitioned between seismic and aseismic slip. The average slip rate $V_F$ of the fault over the duration of the simulation is given by $V_F = V_S + V_A$, where $V_S = \sum M_O/(\mu \ TWL)$ is the seismic slip, due to all the earthquakes rupturing the fault, and $V_A$ is aseismic slip. The average degree of seismic coupling on the fault plane is $\chi = V_S/V_F$ (Scholz, 1998) and represents the proportion of fault slip that occurs by earthquakes and seismic slip. We define the reference fault slip rate as equal to the seismic slip rate of the "only seismic slip" model so that $V_F = V_S \simeq 15$ mm.yr$^{-1}$. This velocity is only given approximatively as the model developed here is stochastic and leads to intrinsic variability in the number and magnitude of earthquakes for the same parametrization. We follow the paradigm of statistically homogeneous long-term fault slip over the fault. The "only seismic slip" model, with an almost uniform spatial distribution of mainshocks, is therefore in average fully coupled, with $\chi = 1$, while the "seismic and aseismic slip" model, displaying a large change with depth of the distribution of mainshocks, is dominated by aseismic slip with $\chi \simeq 0.25$. In the modeling framework developed here, even a fully coupled fault can display significant spatial variations of fault slip rate. Slip rate on the fault plane of the "only seismic slip" model varies between 11.4 to 18.2 mm.yr$^{-1}$ for an average value of ~15 mm.yr$^{-1}$. However, theses spatial variations are randomly distributed and do not follow any specific pattern (Fig. 4).

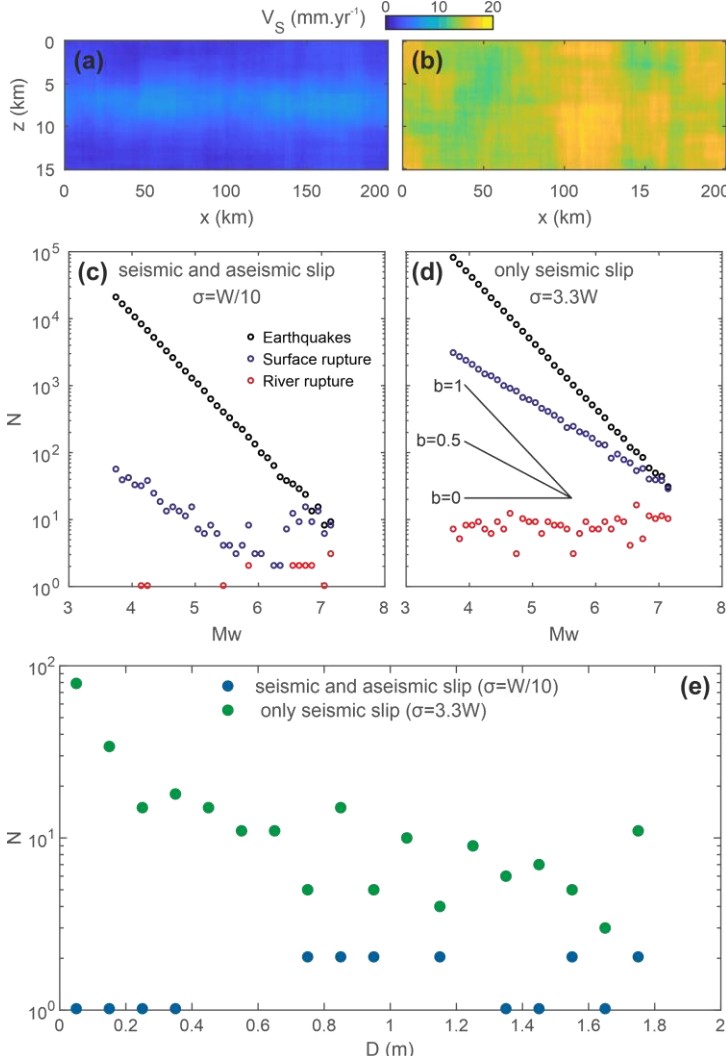

**Figure 4.** Incremental distribution of earthquake magnitude and displacement in surface and at depth. Maps of averaged fault seismic slip rate $V_S$ on the fault plane for a) the "seismic and aseismic slip" and b) the "only seismic slip" models. The scale of the $z$-axis is increased compared to the $x$-axis to enhance readability. c-d) Modeled magnitude distributions of earthquakes on the fault (black circles), earthquakes rupturing the surface (blue circles) and of earthquakes rupturing the river (red circles) for the same models than in panels a and b, respectively. $N$ is here is the incremental number of earthquakes, i.e. $N(m)$. e) Distributions of displacement for earthquake rupturing the river for the considered models, with green and blue circles representing the "only seismic slip" and the "seismic and aseismic slip" models, respectively.

## 3.6 River uplift

A virtual river, orientated orthogonally to the fault trace, crosses the fault trace at its center, at $x = L/2$. This river witnesses the distribution of co-seismic and aseismic displacement modifying its topography and slope. For the sake of simplicity, 1) we assume that any surface rupture generates displacement only in the vertical direction and 2) that co-seismic and aseismic

deformation lead to a block uplift of the hanging wall, homogeneous along the river profile. These 2 assumptions clearly neglect the influence of the fault dipping angle and of the spatial distribution of uplift in surface during an earthquake, which depends mainly on earthquake magnitude, depth, geometry and on the crustal rheology. In turn, earthquakes that do not rupture the surface at the location of the river have no effect on river topography and slope in this simple model. The rate of uplift is equal to $V_F$ at the intersection between the fault trace and the river.

### 3.7 River topographic evolution

To model river erosion, we consider a simple model considering that knickpoints migrate upstream at constant velocity $V_R$ along the $y$-axis, that is perpendicular to the fault trace orientated along the $x$-axis. We show in the Appendice A that this constant migration velocity model corresponds to a prediction of the stream power law (Howard and Kerby, 1983; Howard, 1994; Whipple and Tucker, 1999; Lague, 2014) which holds if drainage area is about constant over the region of interest. Our model is therefore appropriate to model knickpoint migration in near-fault conditions and for large drainage areas. In the following, we only consider the migration of slope patches over short distances upstream, during the $T=10$ kyr of the simulation. We set the horizontal retreat rate to $V_R = 0.1$ m.yr$^{-1}$, which corresponds to a high rate of knickpoint retreat over geological time-scales ($>10^3$ yr) but a moderate one over shorter time-scales (e.g. Van Heijst and Postma, 2001).

### 3.8 Numerical implementation

Numerically, we solve in 2D the evolution of a river profile crossing a fault, subjected to slip during earthquakes and to aseismic slip. After having set the parameters, the model 1) generates mainshocks and aftershocks, including their magnitude, location and timing, and 2) computes the time evolution of the river profile subjected to uplift and erosion. Time stepping combines a regular time step, to account for uplift by aseismic slip, with the time of occurrence of each earthquake rupturing the surface at the location of the river, to account for co-seismic slip. During each aseismic time step, one node of coordinates ($h = 0, y = 0$) is added to the river profile at the downstream end of the river (i.e. the location of the fault trace). During each co-seismic time step, two nodes of coordinates ($h = 0, y = 0$) and ($h = D, y = 0$) are added to the river at the downstream end of the river, to represent the vertical step associated to the co-seismic knickpoint. The remaining nodes, located upstream, are uplifted following the aseismic uplift rate $V_A$ and potential co-seismic displacement. River erosion is accounted for by horizontal advection of river nodes following a constant velocity $V_R$ along the $y$-axis. As we neglect the contribution of horizontal displacement due to fault slip, we do not consider any horizontal advection induced by tectonics, contrary to some previous studies (Miller et al., 2007; Castelltort et al., 2012; Thieulot et al., 2014; Goren et al., 2015).

# 4 Magnitude, displacement and temporal distributions of earthquakes and co-seismic knickpoints

## 4.1 Magnitude distributions

We first use this model to investigate the distribution of earthquake magnitudes that rupture 1) the fault, 2) the surface and 3) the surface at the location of the river (Fig. 4). For clarity, the frequency-magnitude distributions are shown as incremental distributions $N(m)$ and not as cumulative distributions $N(\geq m)$. Unsurprisingly, the frequency-magnitude distribution of earthquakes on the fault follow a negative power-law distribution with an exponent $b = 1$, following the imposed Gutenberg-Richter distribution. Increasing the degree of seismic coupling $\chi$ only shifts the distribution vertically by increasing the total number of earthquakes.

The distribution of earthquakes rupturing the surface follows a negative power law with an exponent -0.5 for the "only seismic slip" model with a high degree of seismic coupling. In the case of the "seismic and aseismic slip" model, characterized by a lower degree of seismic coupling, the distribution follows a more complex pattern. Below a threshold magnitude, here around 6, the distribution follows a negative power law with an exponent -0.5. Above this threshold magnitude, the distribution rises to reach the Gutenberg-Richter distribution and then decreases following the trend of the Gutenberg-Richter distribution. This results from the non-uniformity of the distribution of earthquakes with depth. In this model, large magnitude earthquakes can rupture the surface, without requiring their hypocenters to be at shallow depth. Whereas, small magnitude earthquakes can only rupture the surface if their hypocenters are located close to the surface, which is unlikely due to the shape of the depth-distribution of mainshocks (Fig. 3). The threshold magnitude depends on the depth-distribution of mainshocks, and particularly on its upper limit, but also on the aftershock depth-distribution that extends the range of possible depths due to Omori's law in space.

The distribution of earthquake magnitude rupturing the river follows a uniform distribution for the "only seismic slip" model. This novel result has potentially large implications as it means that a river has an equal probability of being ruptured by large or small earthquakes. This homogeneous distribution results from considering earthquake ruptures at one location and is yet consistent with a Gutenberg-Richter distribution of magnitudes along the modelled 2D fault plane. However, for the "seismic and aseismic slip" model, mostly large-magnitude earthquakes manage to have ruptures cutting the river profile. Low-magnitude earthquakes, except for a few events, do not rupture the river. The magnitude threshold for river rupture is close to 6, similar to the one observed for surface ruptures. To date, there is no universal model of the depth-distribution of earthquakes and of the partitioning between aseismic and seismic slip at shallow depth for intra- or inter-plate faults (e.g. Marone and Scholz, 1988; Scholz, 1998, Schmittbuhl et al., 2015; Jolivet et al., 2015). Yet, our results, i.e. a uniform distribution of earthquake magnitude cutting the river in the fully seismic case or only large magnitude earthquakes rupturing the river for the model dominated by aseismic slip at shallow depth, clearly offer a guide to analyze river profiles in terms of fault properties.

### 4.2 Displacement distributions

Fault displacement $D$ during an earthquake scales linearly with seismic moment $M_O$ (Wells and Coppersmith, 1994; Leonard, 2010), that is related to magnitude by a logarithmic function, $Mw = 2/3 \log_{10}(M_O) - 6.07$ (Kanamori, 1977). It results that a uniform distribution of earthquake magnitude, that is observed for earthquakes cutting the river in the case of the "only seismic slip" model, should lead to a negative exponential distribution of earthquake displacements. The same finding with the numerical model (Fig. 4e). In the case of the "seismic and aseismic slip" model, it is more difficult to quantitatively characterize the resulting distributions due to the lack of events, but we observe a relatively uniform distribution of surface displacements.

### 4.3 Temporal distributions

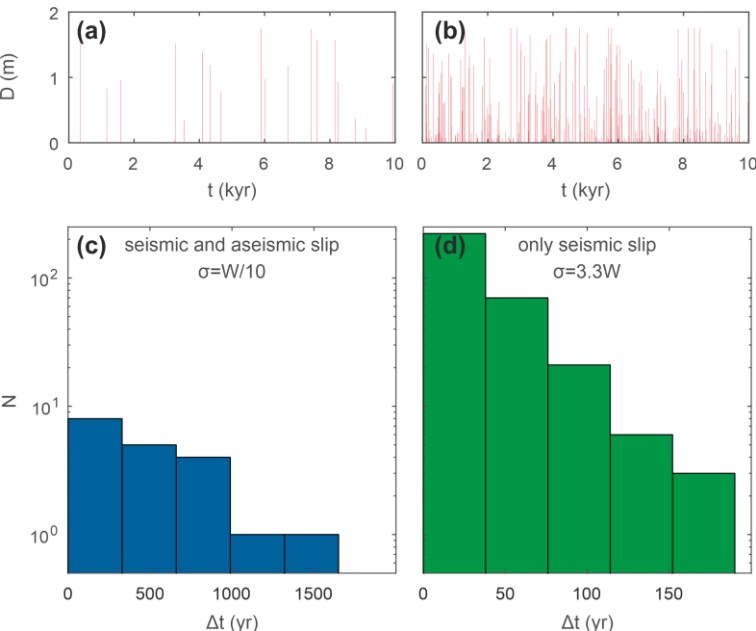

**Figure 5.** Time distribution of earthquakes rupturing the river. a-b) Co-seismic displacements $D$ at the location of the river as a function of time $t$ for each model. c-d) Distribution of inter-event time $\Delta t$ of earthquakes rupturing the surface at the location of the river.

We now investigate the time distribution of earthquakes rupturing the surface at the location of the river and their associated displacement. The "seismic and aseismic slip" and the "only seismic slip" models have 20 and 299 earthquakes cutting the river, respectively. Their average co-seismic displacement is 1 and 0.5 m, respectively. This illustrates that models dominated by aseismic slip have less frequent earthquakes cutting the river, but that their average displacement is greater, due to the censoring of surface ruptures associated to low-magnitude earthquakes (Fig. 4).

Consistent with this last result, the inter-event time $\Delta t$ in-between successive earthquakes cutting the river increases significantly from the "only seismic slip" model to the most "seismic and aseismic slip" model. In other words, the frequency

of surface rupture is higher in the most seismic models and decrease with aseismic slip. This inter-event time distribution follows for each model an exponential decay (Fig. 5), which is consistent with a Poisson process. For the "seismic and aseismic slip" model, the low number of events, 20 earthquakes, precludes characterizing a negative exponential distribution. This exponential decay implies that fault properties have no major effect on the temporal structure of earthquakes cutting a river, only on their frequency.

## 5 Knickpoints along single river profiles

### 5.1 Constant knickpoint velocity

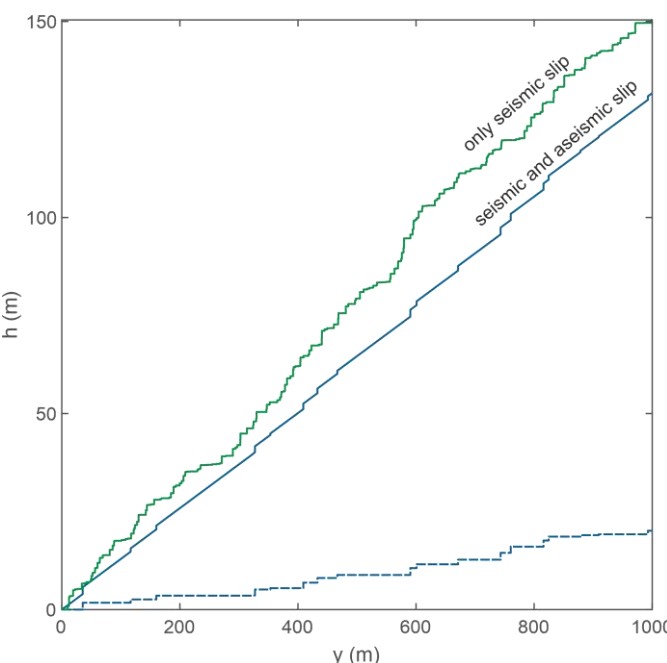

**Figure 6.** Modeled river profiles considering the "only seismic slip" (green line) and the "seismic and aseismic slip" (blue line) models. For this latter, the contribution of seismic slip is showed (dashed blue line).

If the slope patches generated by differential motion across the fault do not migrate horizontally, due for instance to a lack of erosion, the succession of earthquakes would progressively build a vertical fault scarp in this model. Here, we rather consider the case of a migrating topography due to river backward erosion following a kinematic model with $V_R = 0.1$ m.yr$^{-1}$. It results in an averaged river slope just upstream the fault trace of $\varphi = V_F/V_R = 0.15$ or $8.5°$, with $V_F = 15$ mm.yr$^{-1}$ (see Appendix A and Fig. A1). River profiles are obtained for the two different models (Fig. 6). We first only consider seismic slip, so that only earthquakes rupturing the river contribute to topographic building. After $T = 10$ kyr of model duration, the models have resulted in about 20 to 150 m of topographic building for the "seismic and aseismic slip" and "only seismic slip" models,

respectively. The local ratio between $V_S$ and $V_F$ can depart from their fault-averaged values $\chi$, due 1) to the non-homogeneous distribution of co-seismic slip on the fault for models with significant aseismic slip and 2) to the stochasticity of each model. For instance, the "seismic and aseismic slip" model shows an apparent ratio of $20/150 \simeq 0.13$ compared to its average value $\chi = 0.25$. Each successive co-seismic knickpoint is separated by a flat river section, due to the absence of slope building by aseismic slip. As expected, the "only seismic model" displays a larger number of co-seismic knickpoint than the aseismic model. Adding aseismic slip leads to sloped reaches between each knickpoint, (Fig. 6) with slopes equal to $V_A/V_F$. There is obviously a larger slope variability in the models dominated by seismic slip due to a larger number of knickpoints.

## 5.2 Knickpoint velocity that depends on knickpoint height

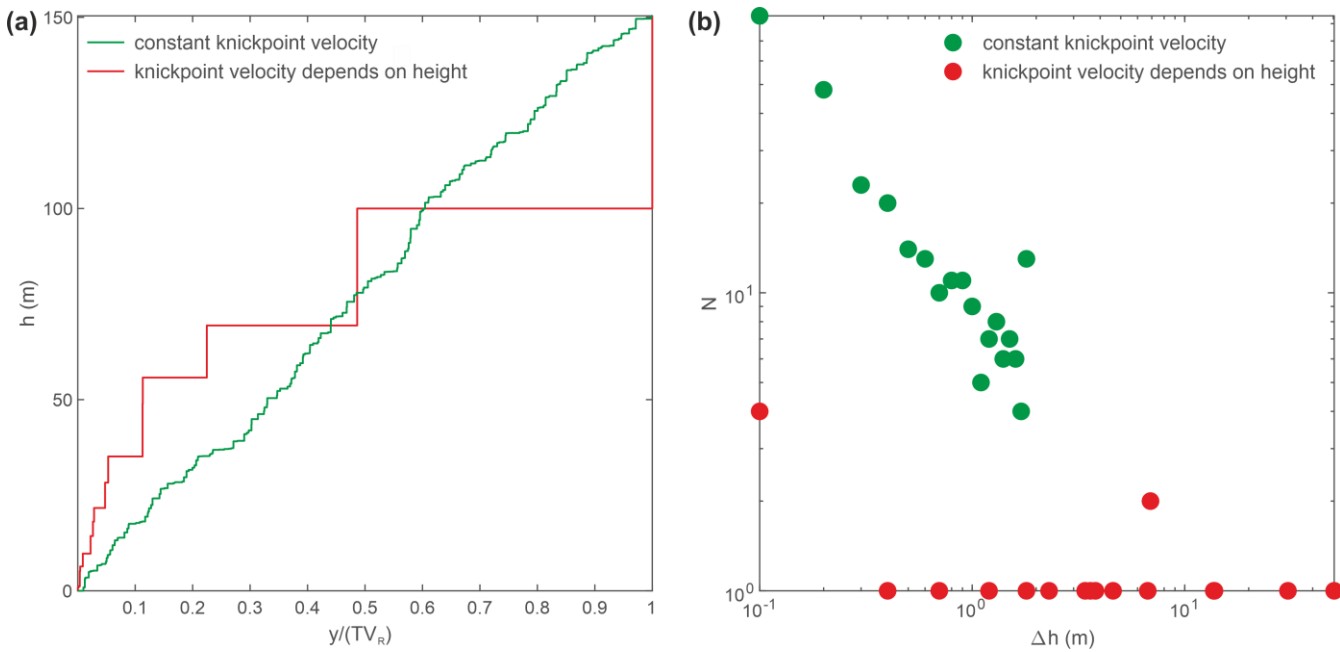

**Figure 7.** Modeled a) river profiles and b) knickpoint height distribution considering a constant knickpoint velocity (green line and circles) or a velocity depending on knickpoint height (red line and circles) .

Even if most simulations of this paper are done with a simple kinematic model using a constant knickpoint velocity, we now consider a model with a knickpoint velocity that depends on knickpoint height with $V_R = r(1 + \Delta h/\Delta h_0)^q$, where $d$ is a constant, set to the previously used constant knickpoint retreat rate 0.1 m.yr$^{-1}$, $\Delta h$ the knickpoint height, $\Delta h_0 = 1$ m a reference knickpoint height and $r = 0.1$ an exponent representing the sensitivity of knickpoint velocity to knickpoint height. This model is motivated by mechanical arguments suggesting a dependency of knickpoint velocity to their height (Scheingross and Lamb, 2017). We allow a quicker knickpoint of height $\Delta h_i$ that encounters a slower knickpoints of height $\Delta h_j$ to merge, forming in turn a single knickpoint of height $\Delta h_i + \Delta h_j$ and of greater speed than the former knickpoints. The resulting river profile can

be compared to the one obtained with the "only seismic slip" model (Fig. 7a). The dependency of knickpoint speed to height leads to a river profile with high but seldom knickpoints. The inter-distance between successive knickpoints increases with total retreat. Small knickpoints only survive close to the fault before being "eaten" by quicker and higher knickpoints during their retreat. Only the highest knickpoints, reaching tens of meters of height, survive after a significant distance of retreat. This behavior is also evidenced when comparing the distributions of knickpoint heights for these two models (Fig. 7b). The dependency of knickpoint velocity to height leads to very few knickpoints, with however a large proportion of them having a metric or decametric scale. This highlights that even limited non-linearities in the knickpoint retreat model can lead to river profiles with significant differences.

### 5.3 Sediment cover, fault burial and knickpoint formation

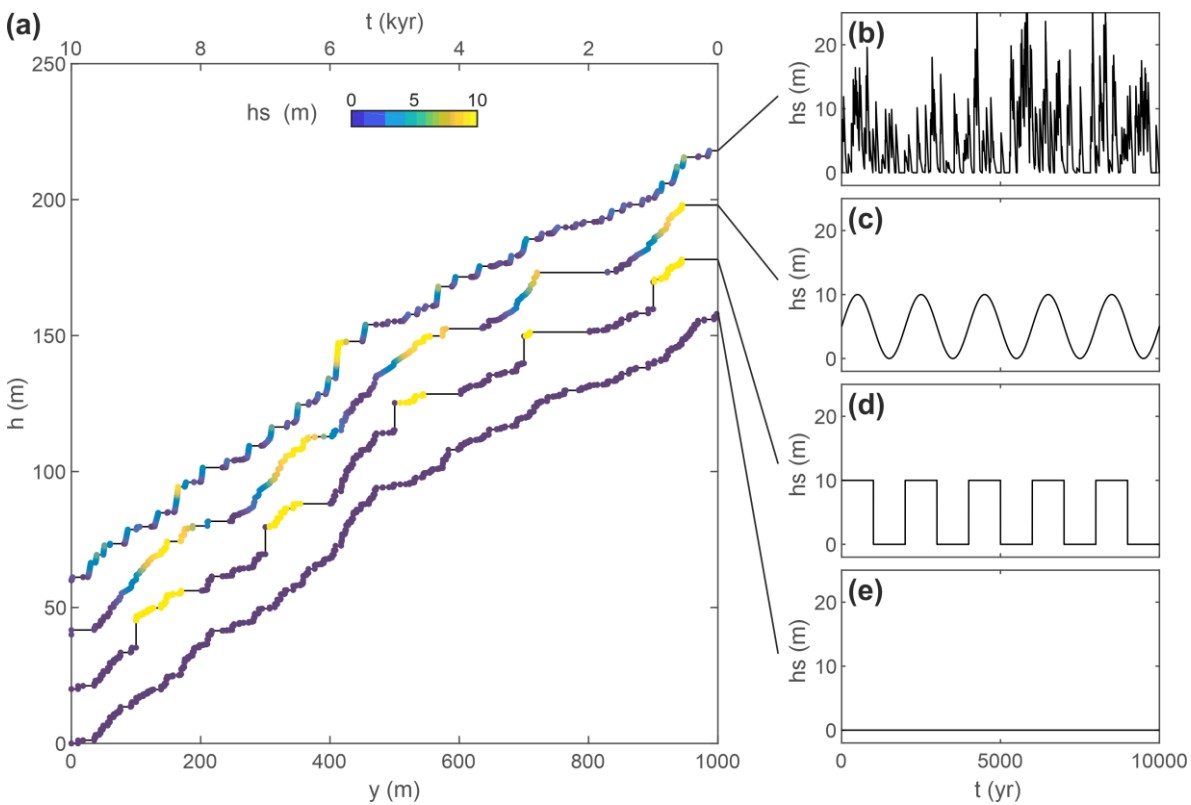

**Figure 8.** Impact of fault burial by sediment cover on river profile. a) River profiles are generated with no sediment cover ($hs = 0$, see panel e), with step-like temporal variations for sediment cover with a periodicity of 2 000 yr (see panel d), with sinusoidal temporal variations for sediment cover with a periodicity of 2 000 yr, mimicking climatic changes (see panel c), with a temporal variation of sediment cover induced by earthquakes (see panel b). The mean sediment cover thickness, $hs$, is equal to 5 m in b, c and d. River profiles are indicated with black lines and the sediment cover thickness at the time of knickpoint formation is indicated by the color of the points. For readability issue, the river profiles are shifted by 20 m on panel a.

We have neglected up to now the role of sediments and their impact on knickpoint formation. More specifically, fault scarps can remain buried during the aggradation phase of an alluvial fan located immediately downstream of the fault (e.g. Carretier and Lucazeau, 2005). This mechanism is suggested to be a primary control of knickpoints and waterfalls formation by allowing the merging of several small co-seismic scarps formed during burial phases into single high-elevation waterfalls that migrate during latter incision phases (Finnegan and Balco, 2013; Malatesta and Lamb, 2018). We test this mechanism and its impact on river profiles using a simple description of fault burial by sediment cover (Fig. 8). At each time step, the formation of a knickpoint can only occur if the fault scarp height, $h(y = 0)$, is greater than the sediment thickness of the alluvial fan, $hs$. In this case, the formed knickpoint height is simply $h(y = 0) - hs$.

Temporal variations of sediment thickness are prescribed using 4 scenarios: 1) no sediment cover, $hs = 0$, corresponding to the reference model (Fig. 8e); 2) a square wave (or step-like) function with a periodicity of 2 000 yr and a maximum amplitude of 10 m (Fig. 8d); 3) a sinusoidal function with a periodicity of 2 000 yr and a maximum amplitude of 10 m (Fig. 8c); and 4) an earthquake-driven sediment cover, where sediment increase instantaneously after each earthquake that rupture the river with an amplitude arbitrarily defined proportional to $(Mw - 5)^2$, followed by a linear decrease over 100 yr, following results by Croissant et al., 2017 (Fig. 8b). This last scenario mimics, in a very simplified manner, the potential transient response of an alluvial fan to the observed increase of river sediment load induced by earthquake-triggered landslides (Hovius et al., 2011; Howarth et al., 2012; Croissant et al., 2017). Alternatively, the periodic scenario mimics the potential response of sediment thickness to some climatic cycles. These scenarios are purely illustrative and do not aim at offering an accurate description of the impact of tectonic or climatic changes on sediment cover dynamics. For each scenario, except the one with no sediment cover, the mean sediment thickness is 5 m. For the sake of simplicity, we only consider "only seismic slip" model with the same temporal sequence of earthquakes in each of the 4 scenarios. Knickpoint velocity is kept constant and equals to $V_R = 0.1$ m.yr$^{-1}$.

The square wave model is useful to assess the impact of abrupt changes in sediment thickness. During the phase of a high sediment cover thickness that lasts 2 000 yr, the scarp progressively builds its height until reaching 10 m during successive fault ruptures. Over this period, there is no knickpoint formation while previously formed knickpoints continue to migrate upstream, leading to elongated flat river reaches upstream of the fault. Once the scarp is re-exposed, the following earthquakes generate knickpoints (yellow dots in Fig. 8a), with their individual height corresponding to each associated earthquake displacement. Then, the abrupt transition from 10 m of sediment thickness to no sediment thickness suddenly exposes 10 m or more of fault scarp that forms a migrating knickpoint of elevation much higher that the largest earthquake displacement, i.e. 1.8 m. Then during the 2 000 yr that follow, with no sediment cover, each earthquake rupture generates a new knickpoint (blue dots in Fig. 8), as in the reference model.

The sinusoidal model, mimicking climatic oscillations (Fig. 8c), displays a relatively similar behavior, except that it does not form 10 m high knickpoints during the phase of degradation of the sediment cover. Instead, this phase leads to the formation of "climatic knickpoints" as the rate of decrease in sediment thickness is greater than the rate of scarp building by fault slip. For the exact same reason, the phase of sediment aggradation is characterized by no knickpoint formation and by flat river

reaches. Knickpoint formation and the signature of the river profile are therefore dominated by the climatic signal controlling sediment aggradation-degradation phases rather than by fault slip.

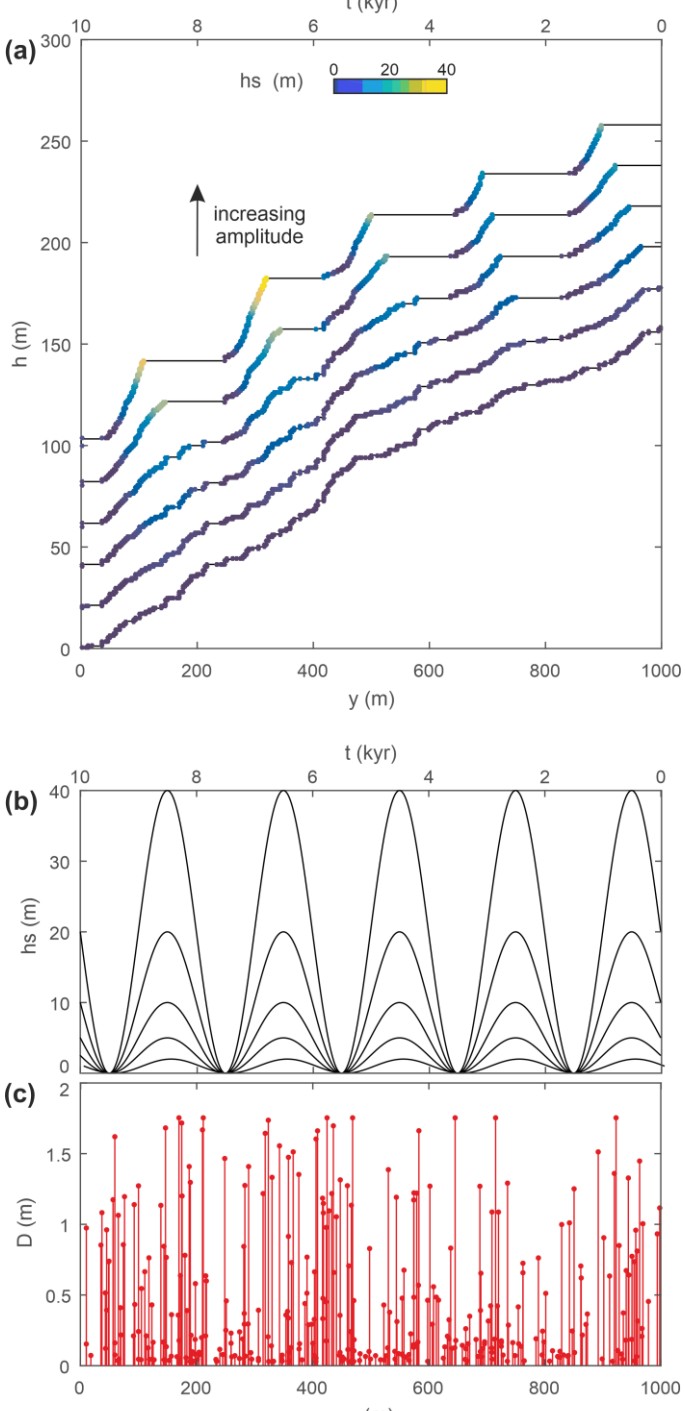

**Figure 9.** Impact of the rate of sediment aggradation and fault burial on river profile. a) River profile simulated with sinusoidal temporal variations for sediment cover, mimicking climatic changes, with a periodicity of 2 000 yr, and an amplitude of 0, 1, 2.5, 5, 10 and 20 m. River profiles are indicated with black lines and the sediment cover thickness at the time of knickpoint formation is indicated by the color of the points. b) Time evolution of the sediment cover $hs$ for the different simulations presented in a. c) Co-seismic displacements $D$ at the location of the river as a function of time $t$ for each model. For a, b and c the x-axis indicates both the distance $y$ along the river and the corresponding time $t$, to relate visually fault displacement, sediment cover and river profile. Time and distance along the river are related through the knickpoint retreat rate $V_R = \frac{y}{t}$.

In these scenarios, the fault-burial mechanism by sediment cover does not necessarily lead to knickpoints with elevation greater than earthquake ruptures, except for abrupt removals of sediment cover such as in the square-wave model. Yet, in all these models, the fault-burial mechanism limits the periods of differential topography building, leading in turn to succession of steepened river reaches or knickzones, corresponding to periods of sediment removal, alternating with low slope river reaches, corresponding to periods of sediment aggradation. Figure 9 illustrates the role of sediment cover in modulating the surface expression of tectonics and co-seismic displacement. For the highest rates of sediment aggradation and removal, river profiles are dominated by the temporal evolution of the sediment cover and not by the activity of the fault. Whereas, for limited sediment aggradation and removal rates, the river profiles and the succession of knickpoints are dominated by the temporal occurrence of earthquakes and not by the temporal evolution of the sediment cover. These results are consistent with the ideas developed by Malatesta and Lamb (2018).

## 6 Knickpoints along successive parallel rivers

### 6.1 From single to several parallel rivers

We now explore the degree of spatial correlation in between the topographic profiles of several parallel rivers flowing across-strike the fault trace. For the sake of simplicity we ignore the role of sediment cover on knickpoint formation and use a constant knickpoint velocity. Paleo-seismological studies using knickpoints to infer fault activity generally consider the distributions of knickpoints along several sub-parallel rivers to lead to statistically robust analyses and to assess the spatial extent of each past earthquake (e.g. Ewiak et al., 2015; Wei et al., 2015; Sun et al., 2016). Correlating topography and knickpoints along the strike of a fault, using parallel rivers, also offers independent means to assess the rupture length and the magnitude of a past earthquake. Using multiple rivers is also less likely to be biased by potential heterogeneities occurring along single rivers. We therefore consider a set of rivers separated by $\Delta x = 1$ km along the strike of the fault, i.e. the $x$-direction. Because 1) the drainage area of each of these rivers can vary by orders of magnitude and 2) because knickpoint retreat rates show a high variability, their knickpoint migration rate $V_R$ is randomly sampled between the range 0.001 to 0.1 m.yr$^{-1}$. Each profile of the 200 rivers share some common topographic characteristic, including their average number of knickpoints and total elevation (Fig. 10). However, their average slopes and the horizontal position $y$ of the knickpoints largely differ due to the variability of

$V_R$. Knowing *a priori* $V_R$ and the duration $T$ of the simulation (i.e. the age of the knickpoints) enables to define a normalized horizontal position relative to the fault, $y/(TV_R)$. Practically, several studies normalized distance by the square root of drainage area, as drainage area is generally used as a proxy for retreat rate (e.g. Crosby & Whipple, 2006). Knickpoints generated at the same time, along different rivers with different retreat rate, share the same normalized distance relative to the fault. This representation is convenient to assess the spatial extent of an earthquake rupturing several rivers along-strike. Non-normalized river profiles are shown on Figure B1.

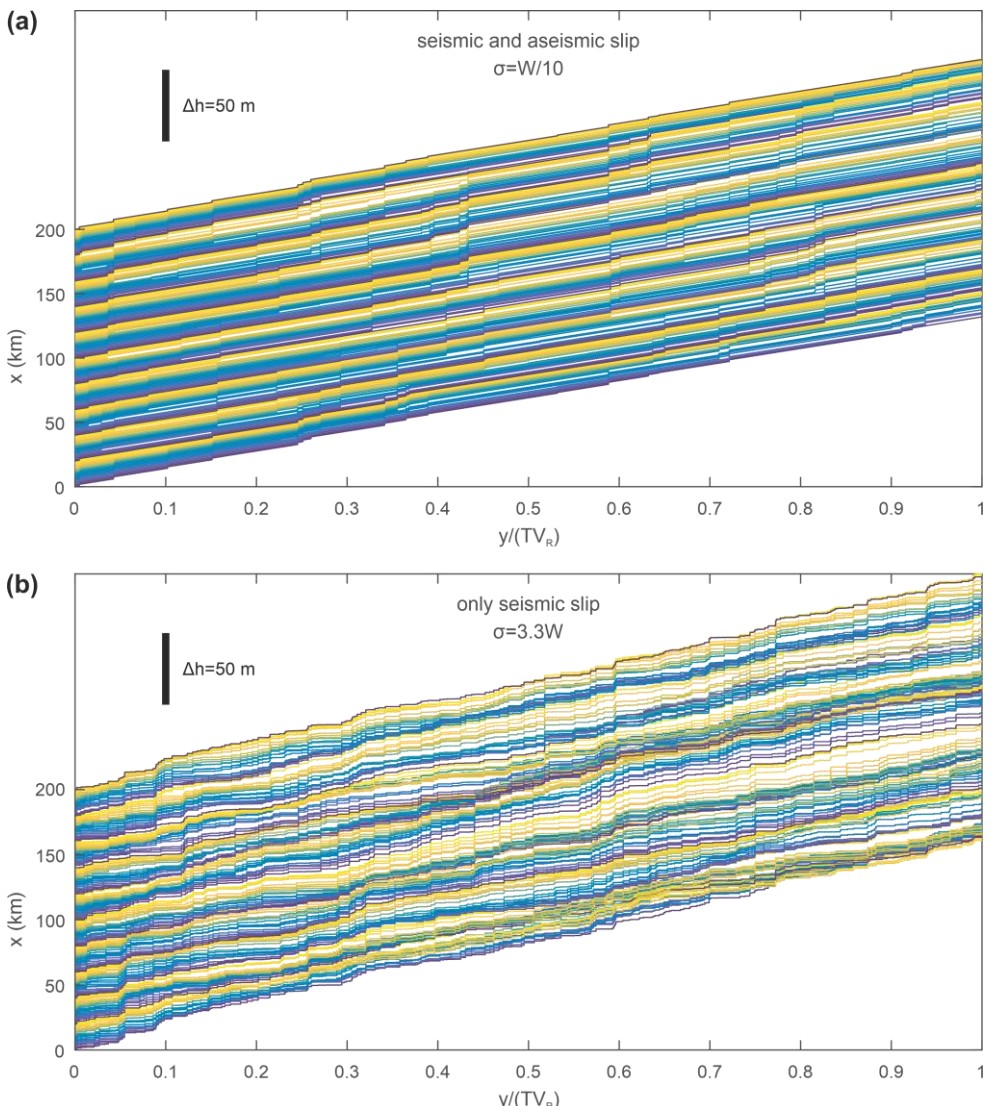

**Figure 10.** Topography of a set of parallel rivers flowing across-strike the fault. a-b) River profiles of 200 rivers separated by 1 km along the strike of the fault, i.e. the x-direction. River elevation $h$ is given along the same axis, with a scaling factor of 1000. River length $y$ across

the strike of fault is normalized by knickpoint migration rate $V_R$ times the duration of the simulation $T$. Non-normalized river profiles are shown on Figure B1. The colorscale is only present to help figure readability.

## 6.2 Knickpoint correlation in-between several parallel rivers crossing the fault

This representation is convenient to assess the degree of correlation of the profiles of the successive rivers. Obviously, there
is no significant topographic correlation when considering rivers with such a high variability in retreat rates, e.g. 0.001 to 0.1 m.yr$^{-1}$. We therefore compute the matrix of correlation between each river elevation profile using the river normalized horizontal distance (Fig. B1). River elevation is corrected or "detrended" from its average slope to remove an obvious source of topographic correlation. We then compute the average coefficient of correlation for a given river inter-distance $\Delta x$ ranging from 0 to 100 km (Fig. 11). The two models, the "only seismic slip" and the "seismic and aseismic slip" models, show a similar
pattern, with a significant positive correlation (>0.5) for rivers separated by less than 14 to 23 km (10 to 45 km if accounting for the standard deviation). The maximum distance over which a correlation is significant corresponds to about 35 km, half the maximum co-seismic rupture length of ~70 km along the considered fault. This illustrates that knickpoints should not be correlated for rivers separated by more than this distance, considering the tectonic setting of this model, and fault dimensions. This correlation distance could increase using a wider fault generating larger magnitude earthquake with longer surface rupture.
We also find that the correlation is better for the model dominated by aseismic slip and showing less knickpoints (Fig. B1). Positive correlations were obtained using horizontal distance normalized by retreat rate. However, using only catchments with similar retreat rates would also lead to positive and significant correlation even when using non-normalized distance.

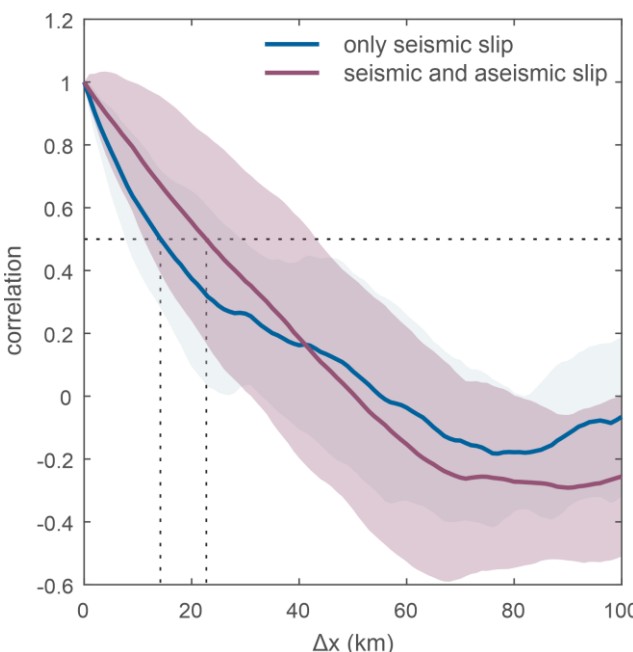

**Figure 11.** Similarity of river profiles along the strike of the fault. Change of the average coefficient of correlation in-between rivers located
along the strike of the fault, with river inter-distance $\Delta x$. The double standard deviation is shown by the extent of the shaded area. In blue,

the "only seismic slip" model, and in purple the "seismic and aseismic slip" model. The average coefficient of correlation and its standard deviation are measured along the diagonals of the correlation matrix (Fig. B1).

## 7 Knickpoint detectability

### 7.1 Knickpoint detectability for the reference model

River profiles are used in many studies to extract co-seismic knickpoints and to assess fault activity and local to regional seismic hazard (e.g. Ewiak et al., 2015; Wei et al., 2015; Sun et al., 2016). It is therefore required to investigate whether modeled knickpoints are statistically detectable. Knickpoint detection often relies on the use of digital elevation models and topographic data (e.g. Neely et al., 2017; Gailleton et al., 2018), which are obtained at a certain scale or resolution. The detectability of each individual knickpoint depends not only on its distance to its adjacent knickpoints, but also on the

horizontal resolution and vertical precision of the topographic data and on the roughness of the riverbed. In the following, we consider that a knickpoint is detectable if its height is greater than the vertical precision of topographic data and if its distance to adjacent knickpoints is greater than the horizontal resolution of topographic data.

Resolutions of topographic data available at the global scale (e.g. SRTM or ASTER) are between 10 and 100 m, with precision not better than a few meters. Local to regional topographic datasets obtained from current airborne Lidar or photogrammetric data or derived from aerial or satellite imagery (e.g. Pléiades) display a resolution between 0.5 to about 1-5 m and a typical

vertical precision of 10 cm above water. Moreover, in the vertical direction, knickpoint detectability depends also on the inherent bed roughness, mean alluvial deposit thickness and the local distribution of sediment grain size. Sediment grains of dimension greater than 0.1 m are commonly found in rivers located in mountain ranges (e.g. Attal and Lavé, 2006), especially at low drainage areas, and there is often a thin layer of sediment covering the channel bed, potentially hiding bedrock features.

If we fully acknowledge the role of river roughness, we here focus on the issue of detectability relative to topographic resolution and precision, for the sake of simplicity, and using knickpoints formed by the "only seismic slip" model.

In terms of vertical precision, a precision of 0.1 m (e.g. Lidar) enables the detection of knickpoints produced by an earthquake as low as magnitude ~4.8 (Fig. 12a). For rivers permanently under water, traditional airborne Lidar using near infrared laser or photogrammetric data cannot measure river bathymetry imposing a detectability level and an uncertainty of knickpoint

height of the order of the water depth. Topographic data with a precision of about 1 m would only enable to detect knickpoints for earthquakes of magnitude above 6.8. It results that about 18 % of the knickpoints are detectable using 1 m of precision, while 72 % are detectable with Lidar data and a precision of 0.1 m (Fig. 12b).  SRTM or ASTER data have precisions of a few meters, at best, that would only enable the potential detection of earthquakes of magnitude ~8 or more.

In terms of horizontal resolution, we assess knickpoint "detectability" by comparing knickpoint inter-distance with the

resolution of topographic data. The distribution of horizontal distance between successive knickpoints (that scales with the distribution of inter-event times) shows that knickpoint inter-distance ranges between less than a millimeter to up to few tens of meters (Fig. 12a). Using a resolution of 10 m, only 7% of the knickpoints can be detected, while a resolution of 1 m increases this percentage to 65 %. Combining horizontal and vertical detectability reduces even more the detectability of knickpoints,

as only 2 or 45 % of the knickpoints are detectable using Lidar (1 m of resolution, 0. 1 m of precision) or DEM (10 m of resolution, 1 m of precision) characteristics, respectively.

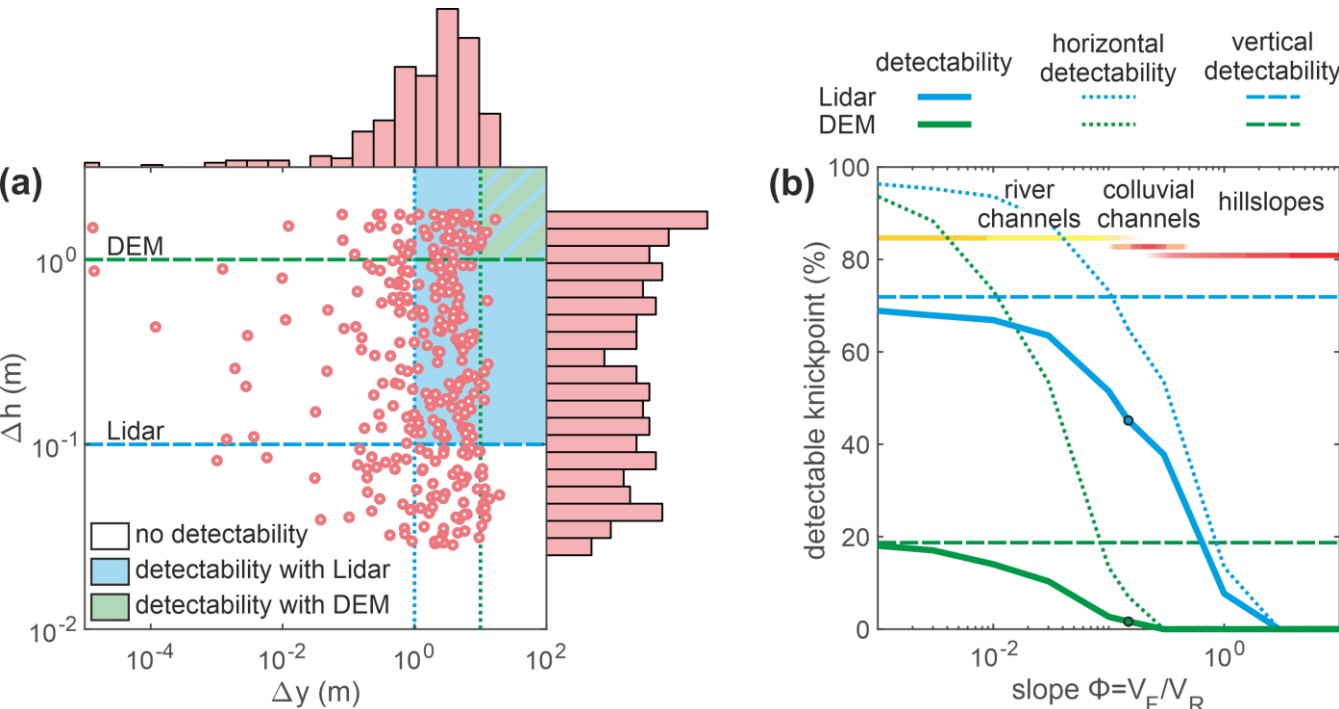

**Figure 12.** Spatial detectability of individual knickpoints. a) Detectability for the "only seismic slip" model of knickpoints (pink circles) depending on their height $\Delta h$ and horizontal distance $\Delta y$. Limits of resolution (dotted lines) and precision (dashed lines) are indicated for DEM (green) and Lidar data (blue). Domains of DEM or Lidar detectability are indicated by plain green or blue colors, respectively. The marginal distributions are indicated by pink bars. b) Full (bold lines), horizontal (dashed lines) and vertical (dotted lines) knickpoint detectability when varying the ratio $\varphi$ between knickpoint retreat rate $V_R$ and fault slip rate $V_F$. $\varphi$ also represents the river slope and the domains of river channels, colluvial channels and hillslopes are indicated by yellow, orange and red bars. Knickpoint detectability is given as a percentage of the number detected knickpoints over the total number of knickpoints. The blue and green dots represent detectability for the model presented in panel a.

## 7.2 Knickpoint detectability along rivers, colluvial channels and hillslopes

We now consider the issue of knickpoint detectability for a broader range of model parameters, in particular fault dimensions, fault slip rate and knickpoint retreat rate. Vertical detectability depends only on the range of considered earthquake magnitude and displacement. As the maximum modeled magnitude is directly limited by the dimension of the modeled fault, considering greater knickpoint requires to extend the dimension of the fault and more specifically its width. In our model, horizontal detectability is directly dependent on the river slope, $\varphi = V_F/V_R = 0.15$. Indeed, the horizontal distance between successive knickpoints increases linearly with knickpoint migration velocity $V_R$, while its decreases linearly with the rate of fault slip $V_F$

that sets the rate of earthquake and knickpoint formation. We here investigate how knickpoint detectability varies with slope $\varphi$ (Fig. 12b). We consider that river channels have a slope below 0.2, colluvial channels between 0.1 and 0.5 and hillslopes above 0.2, following classical slope-area relationships. In terms of horizontal detectability, rivers with a slope $\varphi < 10^{-2}$ have a good detectability, more than 80 % using Lidar or even DEM resolution. Rivers with $10^{-2} < \varphi < 2.10^{-1}$ have a good horizontal detectability using Lidar data, and a moderate one using DEM (10 to 80 %). Colluvial channels have a moderate horizontal detectability using Lidar data, and a bad one using DEM (< 10 %). Hillslopes with $\varphi > 10^0$ have a bad detectability for even with Lidar data. However, the overall detectability of DEM data is below 20 % due to the issue of the vertical detectability, that is low for DEM data. The overall detectability of Lidar data can reach 70% for low values of $\varphi$, in the river domain, while it is moderate or even bad for colluvial channels or hillslopes, respectively. This highlights the need for Lidar data to detect in a more systematic manner knickpoints along river or colluvial channels.

## 8 Discussion

### 8.1 Model limitations

To approach the problem of co-seismic knickpoint formation and their impact on river profile, we have made several simplifying assumptions. The spatial and temporal distribution of earthquakes, including mainshocks and aftershocks, only follow classical statistical and scaling laws. Fault stress state or friction properties, which are first order controlling factors of earthquake triggering (e.g. Scholz, 1998), are not explicitly accounted for. Earthquake ruptures are assumed to be rectangular, to have dimensions scaling with seismic moment and to display a homogeneous displacement (Leonard, 2010), while natural ruptures display more variable behaviors. The relative contributions of seismic and aseismic processes to fault slip, and their spatial distributions are defined in a relatively ad hoc manner. Moreover, co-seismic displacement follows a block uplift mechanism, which contradicts observations and neglects the elasticity of the lithosphere. Yet, it is to be emphasized that block uplift in near fault conditions for large-magnitude earthquakes corresponds to an asymptotic behavior. A more realistic approach is to compute the surface displacement induced by each earthquake using for instance dislocations embedded into an elastic half-space (e.g. Okada, 1985). This alternative approach would also have the benefit of accounting for the surface displacement of earthquakes that do not rupture the surface. Moreover, surface rupture only occurs along a single fault and does not account for off-fault damage (e.g. Zinke et al., 2014), that could also generate knickpoints, or for more complex rupture geometry (e.g. Romanet et al., 2018). Knickpoint retreat along the river profile was modeled using a constant velocity, which corresponds to an asymptotic behavior of the stream power incision model for small migration distance respective to the square root of river drainage area. If the migration of knickpoints or slope patches are classically modeled using the stream power incision model (Rosenbloom and Anderson, 1994; Whittaker and Boulton, 2012; Royden and Perron, 2013), this approach was recently questioned by experimental (Baynes et al, 2018) and field (Brocard et al., 2016) results suggesting no obvious dependency of the migration rate to river discharge. Mechanistic models of waterfall erosion and retreat offer another more accurate but more complex approach (Scheingross and Lamb, 2017).

## 8.2 Model and results applicability to normal and strike-slip faults

The developed model, that was applied in this study to a continental thrust fault, can also be directly applied to a normal fault. Indeed, the adopted scaling relationships between earthquake rupture dimensions or displacements and seismic moment (Leonard, 2010) apply to dip-slip earthquakes and therefore to both normal and thrust faults. The main differences are the polarity of motion between hanging and foot wall, and the dipping angle of the fault. This latter difference vanishes in the developed approach as we assume that rupture displacement occurs only in the vertical direction. Under these limitations and simplifications, all the obtained results in this paper can be therefore directly transcribed to normal faults. Because normal faults tend to have a larger dipping angle, close to 60° in average, than thrust faults, the approximation of purely vertical co-seismic displacement is less incorrect for normal faults. Moreover, strike-slip faults or dip-slip faults can also be accounted for by this model, by simply tuning the parameters of the rupture scaling laws, i.e. $C_1$, $C_2$, and $\beta$. Assuming the depth-distribution of seismicity along different types of faults is identical (a likely incorrect hypothesis), changing the type of fault would not have a major impact on the results presented in this paper.

## 8.3 Knickpoints and horizontal tectonic displacement

Surface ruptures and displacements were only considered in the vertical direction, clearly simplifying the variability in the orientation of natural surface ruptures. If this paper is focused on the vertical expression of fault along river profiles, future work should account for the influence of horizontal tectonic displacement on river profile (e.g. Miller et al., 2007). Accounting for the dip angle of the fault and the associated horizontal tectonic displacement can have two main effects: 1) move knickpoints in the direction of tectonic motion and increase or decrease the apparent retreat rate of knickpoints, in the case of normal or thrust faults, respectively; and 2) move the position of the fault trace through time, as for instance in the case of a thrust sheet when the hanging wall moves over the footwall. We ignore this latter effect and focus on the influence of tectonic motion on knickpoint retreat rate and on river slope. Accounting for the dipping angle of the fault changes the expression of river slope just upstream of the fault. Indeed, fault slip builds topography in the vertical direction at a rate $V_F \sin(\theta)$ while knickpoints retreat by the cumulative effect of erosion and horizontal tectonic displacement at a rate $V_R \pm V_F \cos(\theta)$. The sign $\pm$ is positive for normal faults but negative for thrust faults, as knickpoints are displaced by tectonics towards the fault trace for this latter. It results that the river slope becomes $\varphi = V_F \sin(\theta) / (V_R + V_F \cos(\theta))$ for normal faults and $\varphi = V_F \sin(\theta) / (V_R + V_F \cos(\theta))$ for thrust faults. For rivers, which generally have slopes lower than about 0.1, the horizontal tectonic displacement $V_F \cos(\theta)$ is likely to be negligible compared to the retreat rate by erosion $V_R$, and the slope can be approximated by $\varphi \approx V_F \sin(\theta) / V_R$. However, this approximation does not hold anymore for colluvial channels or for hillslopes as the slope becomes closer to 1. Accounting for tectonic displacement obviously changes the threshold of vertical detectability of knickpoints as their height decreases when decreasing the fault dip angle.

## 8.4 Do mainshocks or aftershocks matter for knickpoints and river profiles?

Aftershocks play a secondary role in the seismicity model considered in this paper. Indeed, for the "only seismic slip" models, aftershocks only represent 18 % of the 442188 earthquakes simulated on the fault. Seismicity is therefore dominated by mainshocks. This is not surprising as about 95% of earthquakes, that follow the Gutenberg-Richter frequency-magnitude distribution, have a magnitude lower than 5 and have in turn a very low probability to generate aftershocks because 1) the aftershock model uses Båth's law with a magnitude difference between any mainshock and their aftershocks, $\Delta Mw = 1.25$, and 2) the minimum magnitude modeled is 3.7. Therefore, aftershocks will only be triggered after intermediate to large magnitude earthquakes, $Mw > 5$, that only represents 5 % of the total number earthquakes. It also results that river profiles in our models are mostly build by mainshocks, and not by aftershocks that only represents 18% of the cumulated uplift. Therefore, developing an aftershock model to include earthquakes and their effects in landscape evolution models represents an additional step in terms of model complexity that is not mandatory. This means that simply accounting for mainshocks by 1) sampling the Gutenberg-Richter distribution to determine earthquake magnitude and 2) randomly sampling their location, already represents a consistent modelling approach towards including earthquakes in landscapes evolution models. Despite that, aftershocks can have significant effects, punctually in time and space, for knickpoint formation, river uplift or even landslide triggering (e.g. Croissant et al., 2019) that justify for some studies the additional complexity of modelling them.

## 8.5 Knickpoint height distribution as a paleoseismological tool?

Co-seismic knickpoints are common geomorphological markers found in seismic areas (Boulton and Whittaker, 2009; Yanites et al., 2010; Cook et al., 2013). Several studies have offered constraints on fault seismogenic activity from the study of river profile and knickpoint height (Boulton and Whittaker, 2009; Ewiak et al., 2015; Wei et al., 2015; He and Ma, 2015; Sun et al., 2016). Natural distributions of knickpoint height are systematically dominated by large heights, corresponding to earthquake magnitudes greater than 5. For instance, the magnitude of earthquakes deduced from knickpoints extracted along rivers crossing the Atacama Fault System, follows a bell shape distribution favoring large magnitude 5.8-6.9 earthquakes (Ewiak et al., 2015). Because the distributions of knickpoints were found to share similarities with the distribution of ruptures directly along the fault scarp, this rules out the hypothesis of fully eroded co-seismic knickpoints generated by small magnitude earthquakes (Ewiak et al., 2015). This observation, of knickpoints dominated by large earthquakes and the censoring of small magnitude earthquakes, is similar to the results obtained in this paper with the model dominated by aseismic slip at shallow depth (Fig. 3e). Alternative explanations for the apparent lack of small knickpoints or scarp ruptures in most natural datasets (Ewiak et al., 2015; Wei et al., 2015; He and Ma, 2015; Sun et al., 2016) include at least 1) the difficulty to detect the limited displacement induced by earthquakes of magnitude 5 or less and 2) the fault burial mechanism (Finnegan and Balco, 2013; Malatesta and Lamb, 2018) that filters out small co-seismic surface ruptures. In any case, the depth-distribution of earthquakes and of their rupture extent exert fundamental control on the resulting height distribution of co-seismic knickpoint.

In turn, our results suggest that knickpoint datasets, that will become more and more accessible thanks to high-resolution topographic data, can be used to assess fault activity. Obviously, the height of knickpoints provide some form of evidence for the earthquakes that have generated them. A negative exponential distribution of knickpoint height points toward a purely seismic fault, while deviations from this trend can suggest aseismic slip or even a slip deficit at shallow depths. The main limitation is yet the poorly known impact of geomorphological processes on evolution of the shape of knickpoints. Some knickpoints along the Atacama Fault System have a reduced height compare to their initial rupture (Ewiak et al., 2015), while some knickpoints produced during Chi-Chi earthquake in 1999 were higher 10 years later (Yanites et al., 2010). These contrasting cases illustrate some potential, and poorly understood, pitfalls in using knickpoints to infer fault and seismic activity.

## 8.6 River dynamics: constant uplift or time-variable uplift with earthquakes?

Most numerical efforts attempting at modeling the long-term (>10-100 kyr) topographic building of mountainous or rift settings have used a constant or smoothly varying uplift rate (e.g. Braun and Willett, 2013; Thieulot et al., 2014; Campforts et al., 2017), not including the variability of uplift rate during the seismic cycle. If using a stream power incision model with a linear dependency to slope $n = 1$, this choice is acceptable as the variability of uplift rate and the associated variability of slope patches shaped throughout the seismic cycle can be averaged out. Moreover, knickpoint retreat rate is in this case independent of slope as this model corresponds to a linear kinematic wave equation, (Rosenbloom and Anderson, 1994; Tucker and Whipple, 2002; Whittaker and Boulton, 2012; Royden and Perron, 2013). However, if using a non-linear dependency of erosion rates to slope, with $n \neq 1$, and only considering a long-term averaged uplift rate, and not its variability, is an approximation that becomes more incorrect with the degree of non-linearity of the model. In other words, the erosion rate of a river profile made of co-seismic knickpoints separated by low-slope river sections built during aseismic periods is not equivalent to the erosion rate of a smooth river profile with the same average slope and built under a constant uplift rate. In a non-linear stream power incision model, the retreat rate is sensitive to slope at a power $n - 1$. For $n > 1$, greater slope patches will migrate quicker than lower slope patches, and vice versa for $n < 1$. While a large proportion of the literature considers the linear stream power incision model (or the unit stream power model) as the reference model, the parametrization of the stream power incision and in particular of the slope exponent $n$ is still an open debate, as is its actual applicability to model knickpoint migration (e.g. Lague, 2014). Moreover, the physics of knickpoint or waterfall retreat likely depends on other variables such as knickpoint height (Holland and Pickup, 1976; Hayakawa and Matsukura, 2003; Haviv et al., 2010; Scheingross and Lamb, 2017), sediment supply (Jansen et al., 2011), lithological structure (Lamb and Dietrich, 2009), and lithological strength (Baynes et al., 2018). Even if this debate is clearly out of the scope of this paper, the implication of this study for the understanding of river erosion and dynamics should not be ignored. Indeed, we have shown that even a slight sensitivity of knickpoint retreat rate to knickpoint height leads to large differences in terms of river profile or knickpoint height distribution, by rapidly merging out all small knickpoints into larger ones associated to greater retreat rates (Fig. 7). Moreover, the modelling results of this study show that the frequency-magnitude distribution of earthquakes rupturing a river is uniform

for purely seismic faults and follows a bell shape, favoring large magnitude earthquakes, for faults with significant shallow aseismic slip. This result offers a complementary - not an alternative – explanation to the fault-burial mechanism (Malatesta and Lamb, 2018) for the apparent larger proportion of high waterfalls.

### 8.7 Co-seismic displacements and knickpoints inside landscape evolution models?

Further implications on the impact of considering earthquakes in landscape dynamics can only be casted by using landscape evolution models (LEMs) (Croissant et al., 2017; Davy et al, 2017; Braun and Willett, 2013; Campforts et al., 2017; Egholm et al., 2011). The developed model in this paper can be implemented in most LEMs to investigate river and landscape response to earthquakes and their successions. However, the main foreseen difficulty is the large variability of inter-event times, that put strong constraints on the time stepping strategy. To overcome this difficulty, a minimum earthquake magnitude can be

defined as a threshold: earthquakes with lower magnitudes are modelled as continuous fault slip, while earthquakes with greater magnitudes are modelled as discrete uplift events during a specific time step. A second difficulty is the spatial discretization of knickpoints that migrate inside the model domain. Most current LEMs use regular grids to discretize surface topography with a uniform resolution. To be consistent with their boundary conditions, such numerical schemes must adapt their spatial resolution to the typical modeled distance between successive knickpoints that can easily go below 1 m (Fig. 7b). This is

problematic as the efficiency of most LEMs scales at best with the number of model nodes (e.g. Braun and Willett, 2013). Using too coarse resolutions would smooth out knickpoints and slope variability leading to similar landscape evolution and dynamics as using a constant uplift, even with non-linear slope dependency. Another more adapted strategy is to use irregular grids, for instance based on Delaunay triangulation, to discretize topography in LEMs (e.g. Braun and Sambridge, 1997; Steer et al., 2011). Despite being less commonly used in LEMs, irregular grids enable to properly account for co-seismic knickpoints

and variable uplift rates by using fine resolutions close to knickpoints and coarser ones in other model domains. This in turn would lead to tractable model durations. Another benefit of irregular grids is their ability to be deformed in the horizontal directions. This is required to account for the horizontal components of co- or inter-seismic displacement that is systematically ignored in LEMs while being of greater amplitude than vertical displacement in convergent or strike-slip settings (e.g. Cattin and Avouac, 2000). Coupling inter- and co-seismic displacement with LEMs represents a future direction to further investigate

the impact of earthquakes and tectonic deformation during the seismic cycle on landscape dynamics. The main remaining limitation is the development of mechanistic models for knickpoint retreat and evolution, a subject that has received recent attention (e.g. Scheingross & Lamb, 2017).

### 9 Conclusions

The accurate modelling of landscape evolution requires accounting for the temporal and spatial variability of surface uplift

and displacement. We propose a statistical model of earthquakes, based on the BASS model (Turcotte et al., 2007), to simulate the slope and height distributions generated by earthquakes and aseismic slip at the intersection between a thrust fault and a

river. The rupture extent and displacement of each earthquake is inferred using classical scaling laws (Leonard, 2010), that can be applied to strike-slip, normal or thrust faults. Slip along the fault plane is partitioned between seismic and aseismic slip using an *ad hoc* spatial distribution of mainshocks along the fault plane. Co-seismic uplift events, with rupture cutting rivers, generate knickpoints that migrate along the river profile following a constant retreat rate.

First, the developed model produces a uniform distribution of earthquake magnitude cutting the river that is obtained while imposing a Gutenberg-Richter frequency-magnitude distribution of earthquakes along the fault plane. In turn, the produced knickpoint heights follow a negative exponential height distribution. The interevent time distribution between successive knickpoints follows an exponential decay.

Second, partitioning shallow slip between seismic and aseismic slip censors the magnitude range of earthquakes rupturing the
surface and cutting the river towards large magnitudes. Poorly coupled faults, dominated by shallow aseismic slip, generate mostly rare and on average high knickpoints while fully coupled faults generate frequent knickpoints of moderate height, in average. Assuming no impact of geomorphological processes on the evolution of the shape of knickpoints, an unlikely hypothesis, these differences in the height distribution of knickpoints offer a guide to assess fault coupling and the shallow partitioning of fault slip over longer-time scales than modern seismology.

Third, our results demonstrate the influence of earthquakes and of fault properties on river profiles. Using a constant knickpoint retreat rate, our simple model produces river profiles made of a succession of flat sections and knickpoints for fully coupled faults, and straight river profiles with a constant slope and few knickpoints for poorly coupled faults. Accounting for a dependency of knickpoint retreat rate to knickpoint height leads to the progressive merging of small knickpoints into larger ones, with a height significantly greater than the vertical offset produced by the largest magnitude earthquakes. Moreover,
fault-burial by intermittent sediment cover can alter the surface expression of fault slip and earthquake activity, when the rate of sediment aggradation/degradation is greater than the rate of fault slip.

Fourth, knickpoint detectability, regarding the horizontal resolution and vertical precision of modern topographic datasets such as Lidar or DEMs, directly depends on the river slope that is equal to the ratio between fault slip rate and knickpoint retreat rate. Decreasing the slope increases the horizontal distance between successive knickpoints and enhance knickpoint
detectability. On the contrary, the vertical detectability is only limited to the vertical precision of topographic relatively to the topographic offsets produces during earthquakes.

Fifth, when considering several parallel rivers distributed along the strike of the fault, a positive correlation between river profiles is obtained if the rivers are separated by less than half than the maximum rupture length occurring on the fault. This correlation is obtained using a horizontal distance normalized by knickpoint migration rates, or when considering rivers with
similar migration rates. The coefficient of correlation becomes significantly positive (>0.5) when the river interdistance is less than about a quarter than the maximum rupture length. For a maximum earthquake magnitude of 7.3, this interdistance corresponds to 14 to 23 km, and does not vary significantly with fault coupling.

Last, the developed model offers insights on the building of slopes and knickpoints by fault activity and earthquakes. This model could also be implemented in landscape evolution models to better infer the role of tectonics and earthquakes on

landscape dynamics. This is pivotal to understand how and why earthquakes build or destroy topography (Parker et al., 2011; Marc et al., 2016), to investigate the feedbacks of erosion on fault dynamics over a seismic cycle (Vernant et al., 2013; Steer et al., 2014) or during orogenesis (Willet et al., 1999; Thieulot et al., 2014), to isolate the feedbacks between river and hillslope dynamics (Valla et al., 2010; Jansen et al., 2011), or to unravel the source-to-sink relationships in seismically active landscapes
(Howarth et al., 2012).

**Code availability.**

A simple Matlab version of the model (Steer and Croissant, 2019) can be accessed through a GitHub and/or a Zenodo repository: https://github.com/philippesteer/RiverFault and https://zenodo.org/record/2654819

**Appendix A: Constant knickpoint retreat rate and the stream power law**

A classical detachment-limited approach to describe the rate of river erosion $E$ is the stream power incision model (Howard and Kerby, 1983; Howard, 1994; Whipple and Tucker, 1999; Lague, 2014) described in Eq. (5):

$$\frac{dh}{dt} = V_F - E = V_F - KA^m \left(\frac{dh}{dy}\right)^n, \tag{5}$$

where $h$ is the elevation of the bedrock bed of the river, $t$ the time, $y$ the distance along the river (i.e. across-strike the fault trace), $S = dh/dy$ the local river slope, $A$ the upstream drainage area, $K$ the erodibility and $m$ and $n$ are two exponents.
Considering a linear dependency of erosion rates to slope, with $n = 1$, the stream power incision model is equivalent to a linear kinematic wave equation. Under this condition, it can be demonstrated (Rosenbloom and Anderson, 1994; Tucker and Whipple, 2002; Whittaker and Boulton, 2012; Royden and Perron, 2013) that knickpoints or slope patches along the river migrate upstream at a rate determined by Eq. (6):

$$V_R = \frac{dy}{dt} = KA^m. \tag{6}$$

Moreover, recent empirical results suggest that using $n = 1$ and $m = 0.5$ is suited to describe knickpoint migration (Lague, 2014). If the total migration distance is small compared to the entire river length, from its source to the modeled frontal thrust fault, the migration velocity $V_R$ can be approximated as a constant. This condition holds only if $KT \ll 1$, considering that river length generally scales with about the square root of drainage area (Hack, 1957). The horizontal knickpoint retreat rate $V_R = 0.1$ m.yr$^{-1}$ can therefore be obtained for an infinity of couples of the $A$ and $K$ parameters, following the relationship $V_R = KA^m$,
that yet must at least satisfy the condition $KT \ll 1$ (Fig. A1). Other conditions exist, including the domain of validity in the $A$ space of the stream power incision model or that the slope generated for a given value of $A$ makes sense in terms of river steepness. However, they are not further considered as the scope of this paper is to develop a general quantitative framework to investigate slope and topographic building by a fault.

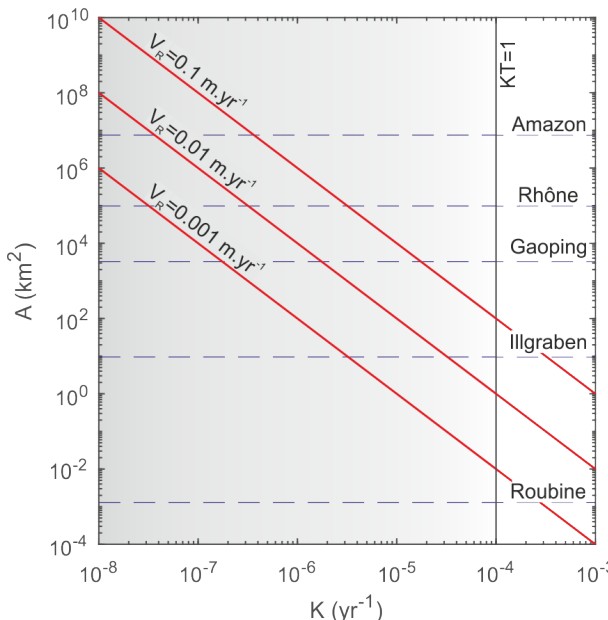

**Figure A1.** Range of possible couples of parameters of river drainage area $A$ and erodibility $K$ for different values of retreat rate $V_R$ (red lines). The vertical black line indicates the uppermost value of $K$, as $KT \ll 1$. The range of acceptable values of $K$ is indicated by a gradient from white (non-acceptable) to grey (acceptable). The drainage area $A$ of some iconic catchments are indicated with dashed blue lines and include the Amazon (south America), Rhône (Europe), Gaoping (Taiwan), Illgraben (Switzerland) and Roubine (France) rivers.

**Appendix B: Parallel rivers topographic correlation**

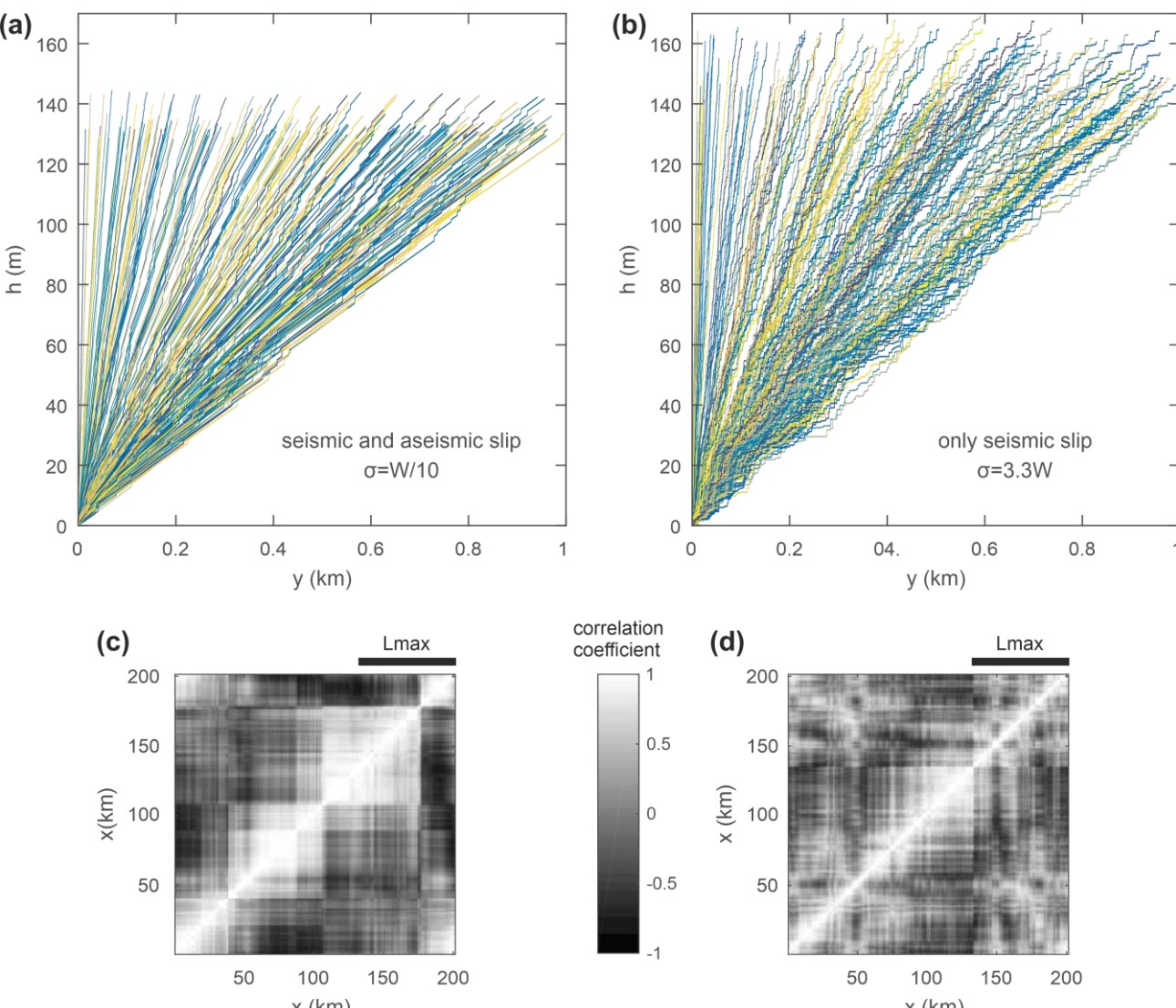

**Figure B1.** Correlation matrixes showing the coefficient of correlation in-between the 200 river profiles shown in Figure 8 a and b, respectively. The correlation is performed on detrended river profiles. Panel a shows results of the "only seismic slip" model, while panel b shows results of the "seismic and aseismic slip" model.

## Appendix C: Table of variable definition and notation

| Variable | Definition |
|---|---|
| $M_0$ | Earthquake moment magnitude |
| $Mw$ | Earthquake magnitude |
| $N(\geq Mw)$ | Cumulative number of earthquakes of magnitude greater than $Mw$ |
| $N(Mw)$. | Incremental number of earthquakes of magnitude $Mw$ |
| $a$ | Gutenberg-Richter earthquake productivity |
| $b$ | Gutenberg-Richter b-value |
| $R$ | Rate of mainshock |
| $\Delta Mw$ | Magnitude difference of Båth's law |
| $p$ | Exponent of the temporal Omori's law |
| $c$ | Offset of temporal Omori's law |
| $q$ | Exponent of the spatial Omori's law |
| $d$ | Offset of the spatial Omori's law |
| $L_{rup}$ | Earthquake rupture length |
| $W_{rup}$ | Earthquake rupture width |
| $C_1$ | Earthquake scaling law constant |
| $C_2$ | Earthquake scaling law constant |
| $\beta$ | Earthquake scaling law exponent |
| $\mu$ | Elastic shear modulus |
| $L$ | Fault length |
| $W$ | Fault width |
| $D$ | Earthquake rupture mean displacement |
| $\theta$ | Fault dip angle |
| $V_F$ | Average fault slip rate |
| $V_S$ | Seismic fault slip rate |
| $V_A$ | Aseismic fault slip rate |
| $\chi$ | Average degree of seismic coupling |
| $\sigma$ | Variance of the normal depth distribution of mainshocks |
| $T$ | Simulation duration |
| $t$ | Simulation time |
| $x$ | Along-fault coordinate |
| $y$ | Along-river coordinate |

| | |
|---|---|
| h | The bedrock riverbed elevation |
| $hs$ | Sediment cover thickness |
| $V_R$ | Knickpoint retreat rate |
| $r$ | Retreat rate constant |
| $\Delta h$ | Knickpoint height |
| $\Delta h_0$ | Reference knickpoint height |
| $q$ | Retreat rate exponent |
| $K$ | Erodibility |
| $A$ | Drainage area |
| $m, n$ | Exponents of the stream power law |
| $\varphi$ | Average river slope just upstream the fault |

**Table C1.** Table of variable definition and notation.

**Author contributions**

PS wrote the paper and designed this study. PS and TC developed the accompanying numerical model. EB and DL motivated the paper through insightful discussions around river profiles and co-seismic knickpoints. All authors checked and revised the text and the figures of the paper, contributed to ideas developed in this study, and discussed the implications for geomorphology and river profile analysis.

**Competing interests.**

The authors declare that they have no conflict of interest.

**Acknowledgments.**

We thank Louise Jeandet, Maxime Mouyen, Michaël Pons, Rodolphe Cattin and Philippe Davy for their helpful comments and for discussions about this work. We also thank Wolfgang Schwanghart, Robert Sare, George Hilley, an anonymous reviewer as well as the editor for their many positive and constructive comments that have contributed to enhance the quality of this manuscript. We acknowledge support by Université Rennes 1, by the Boost'ERC project funded by Région Bretagne, by the French-Taiwanese International Laboratory D3E and by the EROQUAKE project funded by the Agence Nationale de la Recherche (ANR-14-CE33-0005). This project has received funding from the European Research Council (ERC) under the European Union's Horizon Horizon 2020 research and innovation programme (grant agreement No 803721).

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
