# Peer review of "Statistical modelling of co-seismic knickpoint formation and river response to fault slip"

_Earth Surface Dynamics, 2019_

## Referee Comment (RC1) · Anonymous Referee #1 · 11 Mar 2019

Review of Steer et al: "Statistical modelling of co-seismic knickpoint formation and river response to fault slip"

Landscape evolution models (LEMs) are in most cases driven by simplistic boundary conditions. Rock uplift (or translation in the case of strike-slip) due to fault motion may change through time, but is generally treated as a time-averaged rate, and the effects of individual uplift events at the earthquake scale are generally not considered. Yet there is a growing sense in our community that rivers may be valuable indicators of paleoseismic activity, and potentially current seismic hazards. In this contribution, Steer et al make a well-considered and important step towards a better understanding of how rivers respond to sequences of earthquakes. The authors present a simple model for how earthquakes shape river profiles under idealized conditions (i.e., under the

assumption that knickpoints propagate unchanged upstream at a constant velocity). They use the model to make several tests of resulting river profile form, including exploring the effects of 1) the variance of the normal distribution of earthquake depths, 2) along-fault distance of a set of parallel rivers, and 3) various sediment cover scenarios.

The authors make several interesting findings. They find that incorporating both seismic and aseismic slip leads to only larger earthquakes being expressed in the river profile. They find that the degree of seismic coupling (seismic vs aseismic slip) also yields diagnostic changes in profile form. They assess the appropriateness of modern DEMs for extracting knickpoints in different seismic situations, and suggest that in fully coupled faults DEM resolution would have to be unreasonably high to extract knickpoints associated with individual earthquakes. They find a length scale above which correlation between the profiles of parallel rivers along the fault should no longer be expected due to the limited reach of a given earthquake along the fault plane. Finally, they identify situations in which sediment cover dynamics dominate seismically induced knickpoints and vice versa.

GENERAL COMMENTS

I really enjoyed this paper. The study is very novel in considering how the specifics of seismicity are expressed in the river profile. It is also timely as it affects both landscape evolution modelling and the inversion of river profiles obtained from high resolution DEMs. I find the study to be well thought out and well executed, and very appropriate for ESurf. My recommendation is to publish after minor revisions, which have very little to do with changing the science and mostly deal with the presentation. I have four main comments:

1) Explanations of key terms and variables in the seismic modelling portions of the manuscript could be improved. I am a surface processes person, so it may be that I am exceptionally deficient in this department. However, since this paper is submitted to ESurf, I doubt that I am alone in wishing for clearer explanations in the sections

dealing with seismic modelling. For example, the concepts of aseismic and interseismic deformation are not defined where they first occur on line 17 of page 2. I have marked in the specific comments other places where I think some easy additional explanations would help typical ESurf readers understand the paper.

2) These comments pertain to the organization of the manuscript.

The results section is very clear for sections 4.1, 4.2, and 4.3. However, we then jump to "4.4 Knickpoint detectability," which is really not itself a result but an implication of the findings related to knickpoint height and spacing that the authors reported in Figure 6 and section 4.3. To me it seems that section 4.4 and Figure 7 belong in the discussion section. To be clear I think that this discussion of knickpoint detectability is great and should certainly remain in the paper, but since the authors don't actually do any detection of knickpoints in the paper it belongs in the discussion.

Figure 9 shows a very interesting (and important!) result, with major implications for studies using river profiles to infer seismic history. But Figure A1 is important to understanding Figure 9. I suggest that Figure A1 be combined with Figure 9 so that all of this information is in one place in the main text.

Much of section 5.5 describes and contains results of a set of model simulations addressing the role of sediment cover. These are again great, but it does not make sense to me to locate them in the discussion after such sections as "model limitations". Most of section 5.5, including figures 10 and 11, should be relocated to the results section, because these are really just the results of simulations. Any remaining text that does more than describe the results (e.g., the very helpful writing on lines 21-30 of page 23) can remain in the discussion.

3) There are some places in the manuscript where awkward phrasing or sentence structure make reading difficult. I have marked many of these below in "technical corrections," but I would encourage one last thorough proofreading by the authors before resubmission.
4) I was excited to see that the model is available on Github. I ask the authors to consider associating the exact version of the model used for the paper with its own DOI and reporting that DOI in the code availability section. That way, if the model ever gets changed, interested readers can always find the version associated with this paper. This is easily done (∼10 minutes) with GitHub and Zenodo: https://guides.github.com/activities/citable-code/.

SPECIFIC COMMENTS (page.line)

2.17: as noted above, it is important to define seismic deformation, aseismic deformation, and interseismic deformation. If it is too cumbersome for the introduction, use more common wording in the introduction and define the terms in section 2.

2.32: same comment as above for the term "seismic slip"

3.2: Just another clause or sentence here about the BASS model would be good. I know it is explained in detail later, but saying something like "This model uses a standard earthquake sequence model, the branching... "

3.20-3.24: This discussion of Flint's law does not fit into the discussion of earthquake magnitude distributions. It could be moved to section 2.3, or could be eliminated entirely as it is fairly obvious (e.g., if we consider steepness indices rather than slope, there would be no confusion between tectonically driven slope changes and slope change due to decreasing drainage area) and the authors hold drainage area constant in their model anyway.

4.5: define "fast earthquakes"

5.18: There is also a recent field study that should be cited here. Brocard et al (2016) argue that there is poor correspondence between drainage area and knickpoint retreat rate at their sites.

6.12: in the caption for part b, say what squares and circles mean for people reading in grayscale.

7.16: the parameter "chi" in the Figure 2 caption is not defined or explained until page 8, but Figure 2 is referenced on page 7.

8.22: a slightly more thorough explanation of what "seismic coupling" means would be helpful for most ESurf readers.

10.20: I am not sure I would describe the river profile modelling as 2D. Generally the number of coordinates needed to describe a point sets the number of dimensions. I would be tempted to say this is 1D modelling because every point along the profile is associated with an elevation (i.e., no consideration of river width). I understand that where seismically induced knickpoints occur some distances can be associated with two elevation values, but this still seems like a 1D approach, as opposed to for example Croissant et al (2017) in which channel width is also considered.

13.1: "river backward erosion" is awkward. Try an alternate phrase like "upstream knickpoint migration" or something else.

13.11: erodibility is spelled two different ways (also erodability) in this paper. I prefer the first way (and I suspect that the Copernicus typesetters do too).

14.11: as mentioned in the general comments, this subsection seems better suited to the discussion section.

14.19: is there a citation to back up the statement that resolution gets worse in gorges? It seems intuitively true, but a citation would be good.

15.11 and onwards: This is a very nice and important result!

17.1: Add to the caption that panels a/b and c/d correspond to simulations of different variance. I see it in panels a and b, but having it in the caption as well would be helpful.

17.21: as stated in general comments, please consider combining Figure A1 with Figure 9 and keeping it in the main text.

17.26: a novel and important result; this decay in correlation is not often considered by

the surface processes community.

19.13: it would be good to also reference Brocard et al (2016), which provides field evidence compatible with Baynes' experimental result.

21.12: as stated in general comments, please consider moving this to the results section as it is describing an additional set of model simulations.

Figure 11: it might be helpful in panel A to add a label and arrow showing which profile is associated with which amplitude signal. I know it may be fairly obvious, but it would aid interpretation of the figure. Just an arrow with the label "increasing amplitude" pointing in the appropriate direction on panel A would suffice.

TECHNICAL CORRECTIONS

1.21: sub-meter

1.29: interactions among

2.13: builds

2.19: viscous mantle flow

6.16: awkward; consider rewording

7.9: is equal

12.7: typo "nd"

14.18: Resolutions of topographic...

14.22: delete "to"

15.3: last sentence is awkward; rephrase for clarity

16.9 and 10: no hyphen needed

16.10: independent means

17.13: delete "or"

18.9: awkward: rephrase for clarity

20.2: knickpoints were found

21.28: sentence fragment

23.13: a rate? or rates?

23.14: "from no to a large sediment" is awkward

23.17: delete "in turn"

25.13: scales at best linearly with

Caption of Figure A1: two typos: should be "panel a" and "panel b"

REFERENCES

Brocard, G. Y., J. K. Willenbring, T. E. Miller, and F. N. Scatena (2016), Relict landscape resistance to dissection by upstream migrating knickpoints, J. Geophys. Res. Earth Surf., 121, 1182–1203, doi:10.1002/2015JF003678.

---

## Referee Comment (RC2) · Wolfgang Schwanghart (Referee) · 11 Mar 2019

Steer et al. analyse how seismic fault slip translates into the formation of knickpoints along fault-crossing rivers. Combining a stochastic model of earthquakes, earthquake ruptures, and seismic and aseismic slip with a deterministic model of river profile evolution, they present a number of interesting simulation results that can be readily tested in the field or with digital elevation models.

**Major comments**

(1) I really like the comprehensive review of the existing body of literature on knickpoints in river profiles. This is a very helpful resource for anyone who works on this topic. Yet, this entails that the manuscript is quite lengthy at times. I do not consider this as a

significant weakness of the paper, however.

(2) The major part of the methods chapter is concerned with the seismic model. In comparison, the description of the fluvial model is rather terse. This imbalance is also reflected by the code that the authors make available on github. I think that this imbalance arises from the very simplistic treatment of fluvial processes. Please correct me if I am wrong, but isn't the profile just a linear transformation of cumulative slip along the fault? In other words: If knickpoint migration rate is constant in space, then the profile is a linearly scaled timeseries of cumulative slip. Now, constant knickpoint migration rates do make sense in this context, unless we are dealing with catchments of only a few square kilometers area. However, you are raising interesting points in the discussion that you could actually pick up in your study. In particular the nonlinear stream power incision model with exponents unequal to one would be interesting to tackle. Why? Because, if n>1 then knickpoints with larger step heights would travel faster, and potentially coalesce with smaller ones. Now if that is the case, then traces of smaller earthquakes are obliterated by larger ones. This would have severe implications for the inference of fault activity and earthquakes from knickpoints. I am not an expert in Lagrangian numerical models and meshfree simulations, but I guess that the nonlinear model should not be too hard to implement.

Overall, I found the paper a very interesting read. It provides an excellent overview on knickpoint migration and offers an innovative approach to modelling the interaction between fault activity and the fluvial system. However, I think that there is likely a lot to be learnt if running the simulations with the nonlinear stream power incision model, too, and encourage the authors to implement this model.

**ESurfD**

---

## Referee Comment (RC3) · Robert Sare (Referee) · 27 Mar 2019

**Robert Sare (Referee)**

rmsare@stanford.edu

Received and published: 27 March 2019

**GENERAL COMMENTS**

This submission combines a stochastic model of earthquake occurrence with a stream power landscape evolution model to study the spatial distribution and detectability of co-seismic knickpoints generated by a buried thrust fault.

The authors explore the effects of seismic coupling, channel spacing, and exogenous sedimentation on knickpoint expression in isolated and neighboring channels. Among the most significant findings is that a fault with a large seismogenic area and limited aseismic slip may produce a similar number of channel-rupturing earthquakes at all magnitudes. They also identify a channel spacing beyond which parallel channel profiles are poorly correlated due to limited rupture extent, and quantify detection limits for knickpoint identification as a function of data resolution. The authors close with a discussion of the impacts of cyclic sedimentation on knickpoint preservation.

Overall, it is a novel contribution and the methodology is reasonably well documented. I think the manuscript could be accepted after moderate revisions and editing for length and clarity.

**SPECIFIC COMMENTS**

**MAJOR POINTS**

1. In general, the writing could be made more concise. One easy change might be to expand the appendix with some of the details and background. I've tried to indicate sections I felt could be moved to the appendix in the minor points below.

2. A summary table of symbols used would make the study more accessible (included in main text or appendix). This is particularly important because some of the notation ( $\chi$ ) has conflicting uses in geomorphology (drainage-area-normalized channel length) and seismology (seismic coupling coefficient calculated as a ratio of velocities or moments). It would also help to clarify model parameters like  $\sigma$  not always found in physics-based earthquake cycle models, to aid comparison to something like the half width of the seismogenic zone in other studies.

3. A simple schematic figure would be helpful: either a map view of the model domain showing rupture extent and channel spacing, or a profile view of the channel and fault geometry, or both. This may be available in previous work by the authors, in which case a citation pointing to the model set up figure would be very helpful early in the text.

4. I believe the LEM described starting on page 10, line 20 is better described as a one dimensional model of the channel profile. Lateral transport in the y direction is not considered. Regarding this model, a non-linear stream power rule ( $n \neq 1$ ) is worth

**ESurfD**
including in this paper. What if higher slope knickpoints migrate faster than low slope knickpoints? I was surprised to see this discussed at length in Section 5.4 without a comparison of linear and non-linear model results.

5. The methods used in Section 4.4 should be clearly described in the opening paragraph. The analysis counts known co-seismic knickpoints in down-sampled river profiles rather than detecting knickpoints without prior information (which "detectability" might imply to some readers). This provides a useful baseline which should be emphasized early in this section's text. I appreciated the resolution testing summarized in the closing paragraph of this section.

6. The results shown in Figures 10 and 11 are quite interesting and I hope they inspire future work. Could the authors determine something like the maximum detectable tectonic knickpoint height as a function of sedimentation rate and periodicity for tectonically active catchments whose climactic history is well understood? What about superimposing several earthquake cycles with climatically varying sedimentation?

**MINOR POINTS**

1. To justify the Poisson process claim in Section 4.2, a best-fit exponential function and standard error value (or other measure of goodness of fit) could be provided for each of the distributions of inter-event times in Figure 4. It appears that the decay is not necessarily exponential for the most aseismic model (4a). It might be better to weaken this claim if the decay is not exponential in all cases.

2. Figure 5 and parts of Section 4.3 could be moved to the appendix. It is important to justify the choice of VR, but this distracts from the central result in this section.

3. Figure 8 and parts of Section 4.5 could be moved to the appendix as Fig. 9 + A1 and Section 4.5 include sufficient detail.

4. The model fault is a moderately dipping thrust fault, but the knickpoints are generated as vertical discontinuities. How might knickpoint detectability, preservation, and Interactive comment

the profile cross-correlations change if the knickpoints have a finite initial slope or if knickpoints are displaced horizontally? This is addressed in Section 5.1, but it would be particularly interesting in the case of a non-linear stream power rule if knickpoint slopes vary.

5. The link to the GitHub repository is nice to see. For a more direct citation, it would be best to archive the current version of the code and provide a DOI through Zenodo (https://guides.github.com/activities/citable-code/).

**TECHNICAL CORRECTIONS**

I have tried to avoid duplicating editorial points raised by reviewers 1 and 2. I encourage the authors to proofread the revised manuscript.

Page 1 Line 17: "range magnitude" > "magnitude range"

Page 2 Line 25: "following" > "following work"

Page 4 Line 1: "intersect" > "intersection"

Page 20 Line 10: "fundament" > "fundamental"

Page 22 Line 9: "abrupt changes" might be more appropriate than "brutal changes"

**ESurfD**

---

## Referee Comment (RC4) · George Hilley (Referee) · 4 Apr 2019

Review of "Statistical modelling of co-seismic knickpoint formation and river response to fault slip" by Steer et al.

Summary:

This paper combines a stochastic model for releasing accrued moment (by earthquakes, aftershocks, and creep) with a geomorphic model to advect surface-rupturing offsets upstream at a constant velocity along channels. The modeled fault is a 30° dipping thrust fault slipping at 15 mm/yr. The upstream advection rate of offsets generated by earthquakes is set to 10 cm/yr. Using these parameters, the authors investigate the range of generated profile forms, with an eye towards detectability of knickpoints produced by seismogenic offsets and the role that creep plays in river profile development. This speaks to the utilization of the distribution of knickpoint heights as a paleoseismic tool, and the use of constant (or smoothly varying) uplift rate boundary conditions versus individual offset events in modeling river profiles.

Recommendation:

This paper presents an interesting new approach to addressing several important questions in tectonic geomorphology: 1) under what circumstances are profiles approximated by constant or smoothly varying uplift rate conditions versus individual earthquakes, 2) given data sampling and accuracy, under what conditions might river profiles resolve past earthquakes, 3) how does the distribution of surface rupture magnitudes affect the interpretation of knickpoints in river profiles, and 4) what role does creep play in generating the profile forms of rivers traversing faults. I think the work has the potential to provide clarity to many of these issues, at least to the extent that it could nicely frame these problems and lay out a path for future field and modeling studies. As written, the paper contains the seeds of this, but could be reorganized, recast, and extended to be more impactful in this regard. Below, I make several observations and suggestions that are aimed at this purpose. The authors may argue that these suggestions are outside of the scope of this work given their aims. Yet, I worry that in its current form, the paper may not garner the impact it deserves. These are issues of preference, and so I leave it to the authors and editor to decide on whether or not these suggestions are appropriate for the scope of this work. Nonetheless, at a minimum, the paper needs some efficiency improvements and clarifications, which together probably constitute MODERATE TO MAJOR REVISIONS.

General Comments:

1) I have to admit that I found Section 2 confusing and largely unnecessary. I think it is being used to motivate the model that has been developed. But, an alternate approach would be to move directly from the Introduction to the Methods and use the literature

cited to support the model selection. The discussion could then be used to cite and discuss literature that may complicate the model, and what impacts these studies might have on the fundamental results that are reported.

2) I think the presentation of the model in Section 3 could be greatly simplified. It seems as though the following is done: 1) Earthquakes are drawn out of a G-R distribution whose patch sizes and offsets are determined by the Leonard scaling relations (aftershocks are then generated using BASS). 2) The hypocenters of the patches are located according to a Gaussian depth-distribution centered at the midpoint of the model domain, and are uniformly sampled along strike. A uniform slip rate is maintained, such that aseismic creep takes up what the earthquakes do not. 3) From these, surface-rupturing patches are identified, used to uplift surface channel points, and then the profile is advected horizontally at a prescribed rate. I'm not sure there is a need to bring the power-law incision model into this, since it is not really used, to the extent that the advection velocity is assumed constant.

3) I appreciate the comments of the reviewers with regard to examining more flexible geomorphic incision rules. I think that these criticisms are rooted in the fact that the paper is cast in terms of the power-law incision model. As mentioned in (2), the incision model is not really being used here anyway, at least not in any explicit way - the perturbations in the profile are simply being advected headward at a constant velocity. Thus, I would suggest taking the approach that you are using a zeroth-order geomorphic model of constant advection rate to match the simplicity of the offset model (see below for suggestions on this). These limitations can then be discussed in the Discussion, and a path toward future work can be laid out. This avoids issues related to geomorphic model selection, since you would simply be using a kinematically prescribed description of knickpoint creation and migration. The appropriateness of this simplified model could then be discussed later in the paper and put into the context of the incision rule.

4) It seems that with a little more thinking, the spirit of this work could go a long way

in clarifying the role that individual earthquakes play in producing channel profiles. First, I think that the dimensionality of the problem could be reduced and the results could be clarified. The current study uses a single slip rate, which is implemented through events created by the G-R relations, and completed by creep to advect these knickpoints at a velocity, Vr, which is either constant or randomly sampled. It seems as though the profile geometry will be approximated by uniform uplift rate when the spacing between successive knickpoints is small, and will be detectable as individual offset events when this distance is large. This suggests a horizontal length-scale L that varies with Vr * t, where t is the recurrence time of some earthquake. Additionally, the vertical dimension of the model might be normalized by the vertical displacement (d) that occurs during this earthquake. Normalized time in this case would simply reflect the number of earthquakes that have occurred and the fraction of the earthquake cycle that has been experienced. Given this parameterization, if one starts with the simple case of a time-predictable characteristic earthquake, the two length scales can be related through the slip rate along the fault as t = d / (Vs sin(dip)), meaning that L = d (Vr / Vs) / sin(dip). The extremes of this length scale should revert to the single-event, and constant uplift rate cases as t* becomes large.

I understand that this is not what the current study has done, as it is a stochastic model that is used here. But, I wonder if it would be more illustrative to restructure the paper to start with a simple toy model that demonstrates this point. Beginning with a characteristic earthquake that ruptures the surface in this way, one could use the procedure for uplifting and advecting the surface profile in normalized space (x* = x / L; z* = z / d). You could show, in normalized coordinates, that this is just a unity increment in z* with each unity increment in x* at the beginning of each earthquake cycle. When you do this when t* is large, for a range of x* between 0-5, you'll see the earthquakes, when x* is between 0-500, you'll see a constant slope (more-or-less). I think that these are the two end-members that are being sought. Since the uniform uplift case will produce a slope of one in this space, you could even use the deviation from this line as a measure of how far / close to the uniform condition you are. You could then
define a horizontal length-scale that defines the "detection limit" that one might expect to observe in the field, and cast this in terms of the multiple of L at which individual earthquakes bleed into continuous uplift to see those d*(Vr/Vs)/sin(dip) conditions for which one might expect to be able to see individual events clearly.

After this simple exercise is completed, then the stochastic model would be a nice extension of the basic idea. In this case, the productivity rates of the a-value of G-R could be cast in terms of the seismogenic fault slip rates, and this could be used with the maximum magnitude event to define recurrence time, which could serve as the normalizing time-scale. Experiments could be performed, results normalized, etc. to examine the impact of the stochasticity on the character and range of knickpoints generated by such an exercise. Finally, W could then be varied in a way that creep could be added into the analysis, with its impact on the profile form (and knick point detectability) analyzed.

Such an incremental and non-dimensional casting of the problem might help to distill the problem to its essence for illustration, show how increasing realism in the way in which seismic moment is released impacts the profile forms, and broaden the applicability of the analysis to a wide range of fault geometry and slip rate conditions through the non-dimensionalization. The discussion could then be focussed on: 1) How Vr might actually relate to the power-law incision model (since the analysis can be understood without the context of the incision rule), 2) What the impacts of more complicated variations of Vr with things like channel slope might be (i.e., $n \neq 1$), 3) what lithologic / climatic conditions might be appropriate for archiving meaningful paleoseismology information (ranges in K for different watershed areas that produce Vr/Vs ratios that yield detectable knickpoints), 4) what the impact of heterogeneities in lithologies and transport processes (e.g., transport-limited alluvial conditions) might be on profile form and detectability, and 5) other model limitations.

5) I was not altogether clear on how the horizontal motions produced by the 30° dipping thrust fault were actually treated in the knick point evolution model. I think that

the authors are probably resolving the component of slip into the vertical direction, and advecting this offset upstream. This creates two related issues. First, the horizontal motions produced by fault slip will act in an opposing direction to the knickpoint migration, and so some adjustment to Vr needs to be made to account for this. Given Vr and Vs used in the base-case simulation, this will not be a large contributor to the net advection velocity, yet it will constitute punctuated motion during offset events (or constant during creep). Nonetheless, as Vr approaches the 1mm/yr end-member shown in Figure 5, this effect will be important. Related to this, it is worth some text in the discussion that, if the location at which offsets are generated remains fixed, this assumes that the constant-elevation boundary condition is not advected by the horizontal motions. While this is likely the case in many circumstances, I'm not sure it can be regarded as universally true. Advance of the hanging wall of thrusts over the topographic surface in a ramp-flat geometry requires motion of the boundary condition. This will probably happen when Vr - Vs cos(dip) produces a negative value. In these cases, knickpoints will be created and will advance into the hanging wall, but the coordinate system needs to move toward the footwall at this velocity. As the thrust sheet advances over the topography, the ramp-flat geometry might cause motions to parallel to the topographic surface, and so discrete vertical offsets may not exist in these situations as a fault-bend fold develops at the ramp-flat transition. I'm not advocating that the authors implement this, as it is clearly beyond the scope of this work. But, it is an opportunity to frame future studies in the Discussion, which might track down some of these issues.

6) In the spirit of matching the complexity of models to one another, I wonder if the inclusion of aftershocks in the earthquake model creates a mismatch to the constant-advection-velocity geomorphic model. Indeed, I might argue that the appropriate match to the complexity of the geomorphic model that is used would be a time-predictable characteristic earthquake offset. But, I think there is real value in extending the model to creating events according to a G-R distribution. Analyzing the effect of creep also seems important. What was not completely clear was how much of a difference the aftershock component of the model makes in the end analysis. Could it be eliminated

without loss of insight? I am not saying that this is the case, but the process by which the authors build the study – that all of the complexity is introduced at the same time rather than an incremental inclusion of effects to assess each's importance – makes this difficult to determine.

Specific Comments:

I started marking up the PDF in Hypothes.is. Given that the paper may need some restructuring (if the authors find any of the above commentary valuable), I tapered off the detailed editing after Section 2.

https://hyp.is/go?url=urn%3Ax-pdf%3Ace60be1c82ce3c4e151f9f5839f60e0c

Summary:

I hope that the authors find these comments helpful in preparing their revision. Again, I think that this work has the potential to provide clarity as to when, and under what conditions individual offsets may be resolved within the profile of channels. It also has the opportunity to frame many future modeling and field studies focussed on tracking down and field-testing some of the assumptions made in the approach. In this regard, I hope that the authors understand that my comments are aimed at seeing that the work ultimately has the impact that it deserves.

George Hilley.

---

## Author Comment (AC1) · 7 Jun 2019

**Reply to the reviewers**

Dear Editor,

Thank you for your comments and corrections. Please find below our reply in blue font, while reviewer's comments are in black font.

We also wish to thank the four referees for their very positive and constructive comments and for their hard work on our manuscript.

The manuscript has been significantly re-organized and changed according to suggestions and comments by the four referees.

Philippe Steer and co-authors

\*\*\*
* * *
**General comments:**
* * *
I here summarized some significant comments that were common to several referees.

1) Technical corrections, specific comments and proofreading: Referee #1, Robert Sare and George Hilley

Most technical corrections pointed out by the referees were corrected. Note however, that we were unable to access the Hypothesis link given by George Hilley.

2) Paper organization is not optimal as some parts of the Results and Discussion should be switched: Referee #1. Moreover, Referee #1, Wolfgang Schwanghart and Robert Sare have pointed out that the paper is quite lengthy and that the writing could be more concise, in particular by moving minor sections in the appendix.

We agree with this comment and we have therefore reorganized the paper and tried to limit the length of the paper, which was difficult due to the numerous interesting comments made the four referees.

→ For clarity and readability issue, we have separated the Results section in four sections:

- 4 Magnitude, displacement and temporal distributions of earthquakes and co-seismic knickpoints
- 5 Knickpoints along single river profiles
- 6 Knickpoints along successive parallel rivers
- 7 Knickpoint detectability

The part about knickpoint detectability is kept as a result but is appears as the last result. The section on "Fault burial mechanism by intermittent sediment cover" is now part of the Results

→ To reduce (or to not increase) the length of the paper, we have:

- Decided to let Figure A1 in the appendice, despite the suggestion by Referee # 1 to include this Figure in Fig. 9. Indeed, if we agree that Figure A1 can help to understand Figure 9, Figure A1 is not a prerequisite to understand Figure 9.
- Moved some significant parts of the manuscript (including some figures) in the Appendices.
- Shorten some sentences to be more direct.
- Reduced the size of the figures (and of the description of the results) by considering only 2 end-member models instead of the 4 models described in the initial manuscript.

3) Erosion law: Wolfgang Schwanghart and Robert Sare

Citing, for instance, Wolfgang Schwanghart: "However, you are raising interesting points in the discussion that you could actually pick up in your study. In particular the nonlinear stream power incision model with exponents unequal to one would be interesting to tackle. Why? Because, if n>1 then knickpoints with larger step heights would travel faster, and potentially coalesce with smaller ones. Now if that is the case, then traces of smaller earthquakes are obliterated by larger ones. This would have severe implications for the inference of fault activity and earthquakes from knickpoints. I am not an expert in Lagrangian numerical models and meshfree simulations, but I guess that the nonlinear model should not be too hard to implement."

We agree with this comment that implementing a non-linear dependency to slope for the stream power law in a Lagrangian framework is relatively straightforward. However, the main issue is that in this paper we are considering vertical knickpoints, which have by definition an infinite slope. This is not an issue for a linear model, as the slope exponent is n-1=0 (rather than n=1 in the Eulerian framework): $\frac{dy}{dt} = KA^m S^{n-1}$, which is by itself a nice outcome of Lagrangian models. However, for a value of n different than one (i.e. for non-linear models), there is no solution to this equation for infinite slopes. Moreover, we are not confident that larger knickpoints would migrate faster or slower in a non-linear model, for a constant slope.

Yet, the issue of the role of knickpoint height on migration rates is an idea, suggested by empirical (Baynes, personal communcation) and theoretical results, (Scheingross and Lamb, 2017) that deserved to be explored.

→ Therefore, we have added a section in the Results (section 5.2) and a figure (Fig. 7) to explore the role of a dependency of knickpoint retreat rate to their height on river profile development and on knickpoint height distribution.
* * *
**Response to comments by Referee #1:**
* * *
We thank Referee #1 for his work on this manuscript, his positive appreciation and for his many advices.

Landscape evolution models (LEMs) are in most cases driven by simplistic boundary conditions. Rock uplift (or translation in the case of strike-slip) due to fault motion may change through time, but is generally treated as a time-averaged rate, and the effects of individual uplift events at the earthquake scale are generally not considered. Yet there is a growing sense in our community that rivers may be valuable indicators of paleoseismic activity, and potentially current seismic hazards. In this contribution, Steer et al make a well-considered and important step towards a better understanding of how rivers respond to sequences of earthquakes. The authors present a simple model for how earthquakes shape river profiles under idealized conditions (i.e., under the assumption that knickpoints propagate unchanged upstream at a constant velocity). They use the model to make several tests of resulting river profile form, including exploring the effects of 1) the variance of the normal distribution of earthquake depths, 2) along-fault distance of a set of parallel rivers, and 3) various sediment cover scenarios.

The authors make several interesting findings. They find that incorporating both seismic and aseismic slip leads to only larger earthquakes being expressed in the river profile. They find that the degree of seismic coupling (seismic vs aseismic slip) also yields diagnostic changes in profile form. They assess the appropriateness of modern DEMs for extracting knickpoints in different seismic situations, and suggest that in fully coupled faults DEM resolution would have to be unreasonably high to extract knickpoints associated with individual earthquakes. They find a length scale above which correlation between the profiles of parallel rivers along the fault should no longer be expected due to the limited reach of a given earthquake along the fault plane. Finally, they identify situations in which sediment cover dynamics dominate seismically induced knickpoints and vice versa.

GENERAL COMMENTS

I really enjoyed this paper. The study is very novel in considering how the specifics of seismicity are expressed in the river profile. It is also timely as it affects both landscape evolution modelling and the inversion of river profiles obtained from high resolution DEMs. I find the study to be well thought out and well executed, and very appropriate for ESurf. My recommendation is to publish after minor revisions, which have very little to do with changing the science and mostly deal with the presentation.

I have four main comments:

1) Explanations of key terms and variables in the seismic modelling portions of the manuscript could be improved. I am a surface processes person, so it may be that I am exceptionally deficient in this department. However, since this paper is submitted to ESurf, I doubt that I am alone in wishing for clearer explanations in the sections dealing with seismic modelling. For example, the concepts of aseismic and interseismic deformation are not defined where they first occur on line 17 of page 2. I have marked in the specific comments other places where I think some easy additional explanations would help typical ESurf readers understand the paper.

→We now give definitions of some key terms associated to seismic modelling. We now define in the text: seismic slip, aseismic slip, seismic coupling etc.

2) These comments pertain to the organization of the manuscript. The results section is very clear for sections 4.1, 4.2, and 4.3. However, we then jump to "4.4 Knickpoint detectability," which is really not itself a result but an implication of the findings related to knickpoint height and spacing that the authors reported in Figure 6 and section 4.3. To me it seems that section 4.4 and Figure 7 belong in the discussion section. To be clear I think that this discussion of knickpoint detectability is great and should certainly remain in the paper, but since the authors don't actually do any detection of knickpoints in the paper it belongs in the discussion.

Figure 9 shows a very interesting (and important!) result, with major implications for studies using river profiles to infer seismic history. But Figure A1 is important to understanding Figure 9. I suggest that Figure A1 be combined with Figure 9 so that all of this information is in one place in the main text.

Much of section 5.5 describes and contains results of a set of model simulations addressing the role of sediment cover. These are again great, but it does not make sense to me to locate them in the discussion after such sections as "model limitations". Most of section 5.5, including figures 10 and 11, should be relocated to the results section, because these are really just the results of simulations. Any remaining text that does more than describe the results (e.g., the very helpful writing on lines 21-30 of page 23) can remain in the discussion.

We globally agree with this comment (except including Figure A1 inside Figure 9).

→ See response to General comment 2.

3) There are some places in the manuscript where awkward phrasing or sentence structure make reading difficult. I have marked many of these below in "technical corrections," but I would encourage one last thorough proofreading by the authors before resubmission.

Done.

→ See response to General comment 1.

4) I was excited to see that the model is available on Github. I ask the authors to consider associating the exact version of the model used for the paper with its own DOI and reporting that DOI in the code availability section. That way, if the model ever gets changed, interested readers can always find the version associated with this paper. This is easily done (10 minutes) with GitHub and Zenodo: https://guides.github.com/activities/citable-code/.

Excellent advice.

→ The DOI is 10.5281/zenodo.2654819 and we have added a reference: Steer and Croissant, 2019
* * *
**Response to comments by Wolfgang Schwanghart:**
* * *
We thank Wolfgang Schwanghart for his work on our manuscript and for his suggestion to use non-linear erosion laws.

Steer et al. analyse how seismic fault slip translates into the formation of knickpoints along fault-crossing rivers. Combining a stochastic model of earthquakes, earthquake ruptures, and seismic and aseismic slip with a deterministic model of river profile evolution, they present a number of interesting simulation results that can be readily tested in the field or with digital elevation models.

Major comments

(1) I really like the comprehensive review of the existing body of literature on knickpoints in river profiles. This is a very helpful resource for anyone who works on this topic. Yet, this entails that the manuscript is quite lengthy at times. I do not consider this as a significant weakness of the paper, however.

→ See response to General comment 1.

(2) The major part of the methods chapter is concerned with the seismic model. In comparison, the description of the fluvial model is rather terse. This imbalance is also reflected by the code that the authors make available on github. I think that this imbalance arises from the very simplistic treatment of fluvial processes. Please correct me if I am wrong, but isn't the profile just a linear transformation of cumulative slip along the fault? In other words: If knickpoint migration rate is constant in space, then the profile is a linearly scaled timeseries of cumulative slip. Now, constant knickpoint migration rates do make sense in this context, unless we are dealing with catchments of only a few square kilometers area. However, you are raising interesting points in the discussion that you could actually pick up in your study. In particular the nonlinear stream power incision model with exponents unequal to one would be interesting to tackle. Why? Because, if n>1 then knickpoints with larger step heights would travel faster, and potentially coalesce with smaller ones. Now if that is the case, then traces of smaller earthquakes are obliterated by larger ones. This would have severe implications for the inference of fault activity and earthquakes from knickpoints. I am not an expert in Lagrangian numerical models and meshfree simulations, but I guess that the nonlinear model should not be too hard to implement. Overall, I found the paper a very interesting read. It provides an excellent overview on knickpoint migration and offers an innovative approach to modelling the interaction between fault activity and the fluvial system. However, I think that there is likely a lot to be learnt if running the simulations with the nonlinear stream power incision model, too, and encourage the authors to implement this model.

→ See response to General comment 3.
* * *
**Response to comments by Robert Sare:**
* * *
We thank Robert Sare for his work on our manuscript, for his technical comments as well as his suggestion to reorganize some parts of the manuscript.

This submission combines a stochastic model of earthquake occurrence with a stream power landscape evolution model to study the spatial distribution and detectability of co-seismic knickpoints generated by a buried thrust fault. The authors explore the effects of seismic coupling, channel spacing, and exogenous sedimentation on knickpoint expression in isolated and neighboring channels. Among the most significant findings is that a fault with a large seismogenic area and limited aseismic slip may produce a similar number of channel-rupturing earthquakes at all magnitudes. They also identify a channel spacing beyond which parallel channel profiles are poorly correlated due to limited rupture extent, and quantify detection limits for knickpoint identification as a function of data resolution. The authors close with a discussion of the impacts of cyclic sedimentation on knickpoint preservation. Overall, it is a novel contribution and the methodology is reasonably well documented. I think the manuscript could be accepted after moderate revisions and editing for length and clarity.

We thank Robert Sare for his work and his comments on our manuscript.

SPECIFIC COMMENTS

MAJOR POINTS

1. In general, the writing could be made more concise. One easy change might be to expand the appendix with some of the details and background. I've tried to indicate sections I felt could be moved to the appendix in the minor points below.

Done.

→ See response to General comment 2.

2. A summary table of symbols used would make the study more accessible (included in main text or appendix). This is particularly important because some of the notation ($\chi$) has conflicting uses in geomorphology (drainage-area-normalized channel length) and seismology (seismic coupling coefficient calculated as a ratio of velocities or moments). It would also help to clarify model parameters like $\sigma$ not always found in physics-based earthquake cycle models, to aid comparison to something like the half width of the seismogenic zone in other studies.

We agree with this comment.

→ We have added a table of variable notation and definition in Appendice C (to keep the paper not too long).

3. A simple schematic figure would be helpful: either a map view of the model domain showing rupture extent and channel spacing, or a profile view of the channel and fault geometry, or both. This may be available in previous work by the authors, in which case a citation pointing to the model set up figure would be very helpful early in the text.

We agree with this comment.

→ We have added a simple schematic sketch (Figure 1) showing the fault extent, earthquake ruptures as well as a single river profile. For readability issue, it was difficult to display more than one river profile.

4. I believe the LEM described starting on page 10, line 20 is better described as a one dimensional model of the channel profile. Lateral transport in the y direction is not considered. Regarding this model, a non-linear stream power rule (n ≠ 1) is worth including in this paper. What if higher slope knickpoints migrate faster than low slope knickpoints? I was surprised to see this discussed at length in Section 5.4 without a comparison of linear and non-linear model results.

→ See response to General comment 3.

5. The methods used in Section 4.4 should be clearly described in the opening paragraph. The analysis counts known co-seismic knickpoints in down-sampled river profiles rather than detecting knickpoints without prior information (which "detectability" might imply to some readers). This provides a useful baseline which should be emphasized early in this section's text. I appreciated the resolution testing summarized in the closing paragraph of this section.

→ We have added a sentence too explain what we refer to as knickpoint detectability (lines 10-12 page 22): "In the following, we consider that a knickpoint is detectable if its height is greater than the vertical precision of topographic data and if its distance to adjacent knickpoints is greater than the horizontal resolution of topographic data."

6. The results shown in Figures 10 and 11 are quite interesting and I hope they inspire future work. Could the authors determine something like the maximum detectable tectonic knickpoint height as a function of sedimentation rate and periodicity for tectonically active catchments whose climactic history is well understood? What about superimposing several earthquake cycles with climatically varying sedimentation?

We agree that these results are interesting. However, we initially conceived this part as a discussion opening for the paper rather than a detailed result section. The work suggested by Robert Sare is obviously interesting, but we believe it would also lead to a longer paper, which was clearly pointed out as a weakness of the paper by several referees. We also think that this issue probably deserves another more focused paper.

MINOR POINTS

1. To justify the Poisson process claim in Section 4.2, a best-fit exponential function and standard error value (or other measure of goodness of fit) could be provided for each of the distributions of inter-event times in Figure 4. It appears that the decay is not necessarily exponential for the most aseismic model (4a). It might be better to weaken this claim if the decay is not exponential in all cases.

We agree that the most aseismic model is not clearly exhibiting an exponential decay, but this simply results from the lack of events (20 earthquakes) to characterize the distribution. Fitting exponential distribution would only highlight the effect of the number of events, and therefore seems of limited interest.

→ We have added a sentence to explain this (Lines 2-3 page 14).

2. Figure 5 and parts of Section 4.3 could be moved to the appendix. It is important to justify the choice of VR, but this distracts from the central result in this section.

We agree with this comment.

→ See response to General comment 2.

3. Figure 8 and parts of Section 4.5 could be moved to the appendix as Fig. 9 + A1 and Section 4.5 include sufficient detail.

We disagree with this comment as we believe it is important for the readers to see by themselves the similarity of successive rivers along a single fault.

4. The model fault is a moderately dipping thrust fault, but the knickpoints are generated as vertical discontinuities. How might knickpoint detectability, preservation, and the profile cross-correlations change if the knickpoints have a finite initial slope or if knickpoints are displaced horizontally? This is addressed in Section 5.1, but it would be particularly interesting in the case of a non-linear stream power rule if knickpoint slopes vary.

→ See response to General comment 3.

5. The link to the GitHub repository is nice to see. For a more direct citation, it would be best to archive the current version of the code and provide a DOI through Zenodo (https://guides.github.com/activities/citable-code/).

Excellent advice.

→ The DOI is 10.5281/zenodo.2654819 and we have added a reference: Steer and Croissant, 2019
* * *
**Response to comments by George Hilley:**
* * *
We thank George Hilley for his numerous suggestions and insights on our manuscript. They have helped us to analyze with more depth the results of our model. Yet, we were not able to fully accounts for all the comments made by George Hilley, and we hope that the changes we have made to the manuscript answers at least partially to some of them.

Review of "Statistical modelling of co-seismic knickpoint formation and river response to fault slip" by Steer et al.

Summary:

This paper combines a stochastic model for releasing accrued moment (by earthquakes, aftershocks, and creep) with a geomorphic model to advect surface-rupturing offsets upstream at a constant velocity along channels. The modeled fault is a 30 dipping thrust fault slipping at 15 mm/yr. The upstream advection rate of offsets generated by earthquakes is set to 10 cm/yr. Using these parameters, the authors investigate the range of generated profile forms, with an eye towards detectability of knickpoints produced by seismogenic offsets and the role that creep plays in river profile development. This speaks to the utilization of the distribution of knickpoint heights as a paleoseismic tool, and the use of constant (or smoothly varying) uplift rate boundary conditions versus individual offset events in modeling river profiles.

Recommendation:

This paper presents an interesting new approach to addressing several important questions in tectonic geomorphology: 1) under what circumstances are profiles approximated by constant or smoothly varying uplift rate conditions versus individual earthquakes, 2) given data sampling and accuracy, under what conditions might river profiles resolve past earthquakes, 3) how does the distribution of surface rupture magnitudes affect the interpretation of knickpoints in river profiles, and 4) what role does creep play in generating the profile forms of rivers traversing faults. I think the work has the potential to provide clarity to many of these issues, at least to the extent that it could nicely frame these problems and lay out a path for future field and modeling studies. As written, the paper contains the seeds of this, but could be reorganized, recast, and extended to be more impactful in this regard. Below, I make several observations and suggestions that are aimed at this purpose. The authors may argue that these suggestions are outside of the scope of this work given their aims. Yet, I worry that in its current form, the paper may not garner the impact it deserves. These are issues of preference, and so I leave it to the authors and editor to decide on whether or not these suggestions are appropriate for the scope of this work. Nonetheless, at a minimum, the paper needs some efficiency improvements and clarifications, which together probably constitute MODERATE TO MAJOR REVISIONS.

General Comments:

1) I have to admit that I found Section 2 confusing and largely unnecessary. I think it is being used to motivate the model that has been developed. But, an alternate approach would be to move directly from the Introduction to the Methods and use the literature cited to support the model selection. The discussion could then be used to cite and discuss literature that may complicate the model, and what impacts these studies might have on the fundamental results that are reported.

We disagree with this comment as most information given in this state-of-start section is required to introduce modelling strategies and model parameters that are given in the Methods section or later. More importantly, few (or no) papers have synthetized the literature about the links between earthquake activity, knickpoint formation and knickpoint propagation, and we embrace this opportunity in this paper.

2) I think the presentation of the model in Section 3 could be greatly simplified. It seems as though the following is done: 1) Earthquakes are drawn out of a G-R distribution whose patch sizes and offsets are determined by the Leonard scaling relations (aftershocks are then generated using BASS). 2) The hypocenters of the patches are located according to a Gaussian depth-distribution centered at the midpoint of the model domain, and are uniformly sampled along strike. A uniform slip rate is maintained, such that aseismic creep takes up what the earthquakes do not. 3) From these, surface-rupturing patches are identified, used to uplift surface channel points, and then the profile is advected horizontally at a prescribed rate. I'm not sure there is a need to bring the power-law incision model into this, since it is not really used, to the extent that the advection velocity is assumed constant.

We partially agree with this comment.

→ Following this comment, we do not root anymore the theory of our modelling approach in the stream power model and we have therefore removed the stream power law form the Methods section. Instead, we now discuss in Appendice A the relationship between the kinematic model and the stream power law.

→ However, we do not wish to over-simplify the description of the Methods as 1) some details of the methods are required to make the paper self-contained when there is no supporting literature and 2) the methods in itself is novel (at least in the Geomorphology community) and deserves to be precisely described as we hope it could be used in future studies or in other models.

3) I appreciate the comments of the reviewers with regard to examining more flexible geomorphic incision rules. I think that these criticisms are rooted in the fact that the paper is cast in terms of the power-law incision model. As mentioned in (2), the incision model is not really being used here anyway, at least not in any explicit way – the perturbations in the profile are simply being advected headward at a constant velocity. Thus, I would suggest taking the approach that you are using a zeroth-order geomorphic model of constant advection rate to match the simplicity of the offset model (see below for suggestions on this). These limitations can then be discussed in the Discussion, and a path toward future work can be laid out. This avoids issues related to geomorphic model selection, since you would simply be using a kinematically prescribed description of knickpoint creation and migration. The appropriateness of this simplified model could then be discussed later in the paper and put into the context of the incision rule.

See responses to previous comment 2) and to General comment 3

4) It seems that with a little more thinking, the spirit of this work could go a long way in clarifying the role that individual earthquakes play in producing channel profiles. First, I think that the dimensionality of the problem could be reduced and the results could be clarified. The current study uses a single slip

rate, which is implemented through events created by the G-R relations, and completed by creep to advect these knickpoints at a velocity, Vr, which is either constant or randomly sampled. It seems as though the profile geometry will be approximated by uniform uplift rate when the spacing between successive knickpoints is small, and will be detectable as individual offset events when this distance is large. This suggests a horizontal length-scale L that varies with Vr * t, where t is the recurrence time of some earthquake. Additionally, the vertical dimension of the model might be normalized by the vertical displacement (d) that occurs during this earthquake. Normalized time in this case would simply reflect the number of earthquakes that have occurred and the fraction of the earthquake cycle that has been experienced. Given this parameterization, if one starts with the simple case of a time-predictable characteristic earthquake, the two length scales can be related through the slip rate along the fault as t = d / (Vs sin(dip)), meaning that L = d (Vr / Vs) / sin(dip). The extremes of this length scale should revert to the single-event, and constant uplift rate cases as t* becomes large.

I understand that this is not what the current study has done, as it is a stochastic model that is used here. But, I wonder if it would be more illustrative to restructure the paper to start with a simple toy model that demonstrates this point. Beginning with a characteristic earthquake that ruptures the surface in this way, one could use the procedure for uplifting and advecting the surface profile in normalized space (x* = x / L; z* = z / d). You could show, in normalized coordinates, that this is just a unity increment in z* with each unity increment in x* at the beginning of each earthquake cycle. When you do this when t* is large, for a range of x* between 0-5, you'll see the earthquakes, when x* is between 0-500, you'll see a constant slope (more-or-less). I think that these are the two end-members that are being sought. Since the uniform uplift case will produce a slope of one in this space, you could even use the deviation from this line as a measure of how far / close to the uniform condition you are. You could then define a horizontal length-scale that defines the "detection limit" that one might expect to observe in the field, and cast this in terms of the multiple of L at which individual earthquakes bleed into continuous uplift to see those d*(Vr/Vs)/sin(dip) conditions for which one might expect to be able to see individual events clearly.

After this simple exercise is completed, then the stochastic model would be a nice extension of the basic idea. In this case, the productivity rates of the a-value of GR could be cast in terms of the seismogenic fault slip rates, and this could be used with the maximum magnitude event to define recurrence time, which could serve as the normalizing time-scale. Experiments could be performed, results normalized, etc. to examine the impact of the stochasticity on the character and range of knickpoints generated by such an exercise. Finally, W could then be varied in a way that creep could be added into the analysis, with its impact on the profile form (and knick point detectability) analyzed.

Such an incremental and non-dimensional casting of the problem might help to distill the problem to its essence for illustration, show how increasing realism in the way in which seismic moment is released impacts the profile forms, and broaden the applicability of the analysis to a wide range of fault geometry and slip rate conditions through the non-dimensionalization. The discussion could then be focussed on: 1) How Vr might actually relate to the power-law incision model (since the analysis can be understood without the context of the incision rule), 2) What the impacts of more complicated variations of Vr with things like channel slope might be (i.e., n = 1), 3) what lithologic / climatic conditions might be appropriate for archiving meaningful paleoseismology information (ranges in K for different watershed areas that produce Vr/Vs ratios that yield detectable knickpoints), 4) what the impact of heterogeneities in lithologies and transport processes (e.g., transport-limited alluvial conditions) might be on profile form and detectability, and 5) other model limitations.

We appreciate this comment that is very insightful. Yet, we are not convinced that extending our modelling approach to- or normalizing our result by- a time-predictable model is relevant.

First, the notion of time-predictable earthquake was popular in the 80's due to its simplicity and apparent quality to reproduce some paleo-seismological records. However, time-predictable models are generally not considered anymore when modelling more resolute paleo-seismological records or to infer seismic hazards (which are the two common disciplines that use statistical models to investigate spatio-temporal series of earthquakes), and most study now consider mechanical-based model or ETAS. Moreover, time-predictable models are already very routinely used in landscape evolution models (and yet most model developers do not conscientiously know it) that generally consider a constant time-step and uplift rate with a sharp boundary condition, which directly translates into a time-predictable model of block uplift.

Second, the link between a time-predictable earthquake model and the stochastic model we use is difficult to establish because:

1) The notion of characteristic recurrence time is fundamental for a time-predictable earthquake model but has no obvious mathematical meaning in a stochastic model as there is no characteristic magnitude. Recurrence time is generally defined as the (mean) time separating successive events of magnitude above a defined threshold. In the purely seismic model, considering a magnitude threshold of 7 or 7.3 (which is the maximum magnitude allowed using the fault dimension of our model) leads to significatively different mean recurrence time of 240 years (min=3, max=1400 years) or 700 years (min=3, max=2340 years), while the minimum height of the knickpoints formed by these magnitudes, 1.25 or 1.7 m, is similar. Normalizing time (or knickpoint height and horizontal distance) by these two recurrence times (or by the recurrence time multiplied by knickpoint or fault velocity) would lead to very different results.

2) The height distribution of knickpoint has no dependency over fault slip rate. Indeed, increasing fault slip rate will only increases earthquake frequency without changing the distribution or range of earthquake magnitudes. Only a change in fault dimension will result in a change in the upper bound of the range of modelled earthquake magnitudes.

However, we agree that the section on "Knickpoint detectability" could have been described with a less model-dependent formalism. Following this comment, we now assess knickpoint detectability using a range of fault slip rate Vf and knickpoint retreat rate Vr. Indeed, the ratio Vf/Vr, that is a dimensionless number that directly represents the river slope, also conditions the detectability of knickpoints: 1) Increasing Vf increases the frequency of knickpoint formation and decreases the horizontal spacing between successive knickpoints; 2) Increasing Vr increases the spacing between successive knickpoints. With this new analysis, we can now predict (under the model limitations) what would be the detectability of knickpoints along any bedrock river, colluvial channel or hillslope with only requiring knowledge of its slope. As the slope of a channel or hillslope is relatively easy to obtain, this clearly represents a result with potentially large implications.

→ We have therefore deeply modified the section on knickpoint detectability (section 7) and the associated figure (Fig. 12) to account for these modifications.

5) I was not altogether clear on how the horizontal motions produced by the 30 dipping thrust fault were actually treated in the knick point evolution model. I think that the authors are probably resolving the component of slip into the vertical direction, and advecting this offset upstream. This creates two related issues. First, the horizontal motions produced by fault slip will act in an opposing direction to the knickpoint migration, and so some adjustment to Vr needs to be made to account for this. Given Vr and Vs used in the base-case simulation, this will not be a large contributor to the net advection velocity, yet it will constitute punctuated motion during offset events (or constant during creep). Nonetheless, as Vr approaches the 1mm/yr end-member shown in Figure 5, this effect will be

important. Related to this, it is worth some text in the discussion that, if the location at which offsets are generated remains fixed, this assumes that the constant-elevation boundary condition is not advected by the horizontal motions. While this is likely the case in many circumstances, I'm not sure it can be regarded as universally true. Advance of the hanging wall of thrusts over the topographic surface in a ramp-flat geometry requires motion of the boundary condition. This will probably happen when Vr - Vs cos(dip) produces a negative value. In these cases, knickpoints will be created and will advance into the hanging wall, but the coordinate system needs to move toward the footwall at this velocity. As the thrust sheet advances over the topography, the ramp-flat geometry might cause motions to parallel to the topographic surface, and so discrete vertical offsets may not exist in these situations as a fault-bend fold develops at the ramp-flat transition. I'm not advocating that the authors implement this, as it is clearly beyond the scope of this work. But, it is an opportunity to frame future studies in the Discussion, which might track down some of these issues.

The model we have developed does not consider knickpoint advection during fault motion. Moreover, we have simplified our model by imposing that all motion on the fault act in the vertical direction, neglecting the role of the dip angle of the fault.

→ Following this comment, we now discuss in section 8.3 (page 2.5) the influence of horizontal tectonic displacements on our results.

6) In the spirit of matching the complexity of models to one another, I wonder if the inclusion of aftershocks in the earthquake model creates a mismatch to the constant advection-velocity geomorphic model. Indeed, I might argue that the appropriate match to the complexity of the geomorphic model that is used would be a time-predictable characteristic earthquake offset. But, I think there is real value in extending the model to creating events according to a G-R distribution. Analyzing the effect of creep also seems important. What was not completely clear was how much of a difference the aftershock component of the model makes in the end analysis. Could it be eliminated without loss of insight? I am not saying that this is the case, but the process by which the authors build the study – that all of the complexity is introduced at the same time rather than an incremental inclusion of effects to assess each's importance – makes this difficult to determine.

We agree with this comment. Indeed, aftershocks play a secondary role in terms of river uplift or knickpoint formation, while they require a relatively high-level of complexity for their implementation in the model compared to mainshock modelling (i.e. that simply follows the Gutenberg-Richter law).

→ We now discuss the role of mainshocks and aftershocks in section 8.4 of the Discussion.

Specific Comments:

I started marking up the PDF in Hypothes.is. Given that the paper may need some restructuring (if the authors find any of the above commentary valuable), I tapered off the detailed editing after Section 2.

https://hyp.is/go?url=urn%3Ax-pdf%3Ace60be1c82ce3c4e151f9f5839f60e0c

→ We were unfortunately unable to access this link that redirected us to an URL error.

Summary:

I hope that the authors find these comments helpful in preparing their revision. Again, I think that this work has the potential to provide clarity as to when, and under what conditions individual offsets may

be resolved within the profile of channels. It also has the opportunity to frame many future modeling and field studies focused on tracking down and field-testing some of the assumptions made in the approach. In this regard, I hope that the authors understand that my comments are aimed at seeing that the work ultimately has the impact that it deserves.

We have tried to enhance the quality of our paper regarding these insightful comments. We hope we have at least partially succeeded in convincing the referee of this.